# HYPER HAWKES PROCESSES: INTERPRETABLE MODELS OF MARKED TEMPORAL POINT PROCESSES

## ABSTRACT

Foundational marked temporal point process (MTPP) models, such as the Hawkes process, often use inexpressive model families in order to offer interpretable parameterizations of event data. On the other hand, neural MTPPs models forego this interpretability in favor of absolute predictive performance. In this work, we present a new family MTPP models: the *hyper Hawkes process* (HHP), which aims to be as flexible and performant as neural MTPPs, while retaining interpretable aspects. To achieve this, the HHP extends the classical Hawkes process to increase its expressivity by first expanding the dimension of the process into a latent space, and then introducing a hypernetwork to allow time- and data-dependent dynamics. These extensions define a highly performant MTPP family, achieving state-of-the-art performance across a range of benchmark tasks and metrics. Furthermore, by retaining the linearity of the recurrence, albeit now piecewise and conditionally linear, the HHP also retains much of the structure of the original Hawkes process, which we exploit to create direct probes into *how* the model creates predictions. HHP models therefore offer both state-of-the-art predictions, while also providing an opportunity to "open the box" and inspect how predictions were generated.

## 1 INTRODUCTION

In modern machine learning, the pursuit of predictive accuracy often comes at the expense of interpretability. This trade-off is especially pronounced in marked temporal point processes (MTPPs): classical models such as the Hawkes process (Hawkes, 1971) offer transparent parameters but underfit real-world data; while neural models such as the neural and transformer Hawkes processes (Mei and Eisner, 2017; Zuo et al., 2020) achieve state-of-the-art performance but lack clear mechanisms for attributing predictions to specific past events. General interpretability methods for neural networks (Räuker et al., 2023; Chefer et al., 2021; Rai et al., 2024; Maheswaranathan and Sussillo, 2020) tend to be indirect or ambiguous, leaving no MTPP approach that combines strong predictive power with precise, event-level interpretability.

We introduce the *hyper Hawkes process* (HHP), a new family of intensity-based MTPP models designed to close this gap. Our HHP is illustrated in Figure 1. The HHP extends the classical Hawkes process by (i) lifting recurrent dynamics into a latent space, decoupling them from mark dimensionality, and (ii) using a history-dependent hypernetwork (Ha et al., 2017) to adapt dynamics over time conditioned on the event history. These extensions enhance expressivity while preserving the linear recurrence and branching structure of Hawkes processes. We then exploit this for efficient event-level attribution using influence measures inspired by linear regression diagnostics, such as DFBETA and DFFIT (Belsley et al., 2005). Through extensive empirical evaluation on real-world datasets, we find that HHP consistently outperforms both classical and state-of-the-art MTPP models.

**Our main contributions are:**

- We propose the *hyper Hawkes process* (HHP), combining the interpretability of classical models with the expressivity of neural architectures.
- We demonstrate, through extensive experiments, that HHP achieves state-of-the-art or near state-of-the-art performance on real-world MTPP benchmarks.
- We then develop efficient, per-event attribution methods exploiting the structure of the HHP, enabling direct insight into model mechanics, and explore these on synthetic tasks.

Table 1: [Added] Summary of the emperical performance of the hyper Hawkes process (HHP) we introduce in this work. We show model rankings across six different metrics, each an average across seven different real-world datasets [Added] and five randomly initialized models, and an aggregated composite ranking, averaging the per-metric ranks. **Bold** entries correspond to best result, and underlined for second-best. Lower ranks indicate better performance. [Added] Full numerical results for all metrics and models are included in Table 2, Table 5 and Table 6.

| Model | | Time Metrics | | | Mark Metrics | | | Composite |
|-------|--|------------|------|-----|------------|----------|-----|-----------|
| | | Likelihood | RMSE | PCE | Likelihood | Accuracy | ECE | Rank |
| RMTPP | (Du et al., 2016) | 7.0 | 5.1 | 5.6 | 6.9 | 6.6 | 6.0 | 6.2 |
| NHP | (Mei and Eisner, 2017) | 4.3 | 3.1 | 5.6 | 3.3 | 2.3 | 5.0 | 3.9 |
| SAHP | (Zhang et al., 2020) | 5.9 | 4.9 | 5.4 | 7.4 | 7.1 | 7.1 | 6.3 |
| THP | (Zuo et al., 2020) | 7.6 | 4.6 | 5.3 | 6.0 | 6.1 | 5.1 | 5.8 |
| IFTPP | (Shchur et al., 2020) | 2.9 | 5.4 | **2.3** | 4.9 | 5.1 | **1.9** | 3.7 |
| AttNHP | (Yang et al., 2022) | 4.1 | 6.7 | 4.6 | 3.3 | 3.6 | 3.6 | 4.3 |
| S2P2 | (Chang et al., 2025) | **1.7** | 2.6 | 3.4 | 2.4 | 2.4 | 3.4 | **2.7** |
| **HHP** | **(Ours)** | 2.6 | **2.3** | 3.9 | **1.9** | **1.9** | 3.7 | **2.7** |

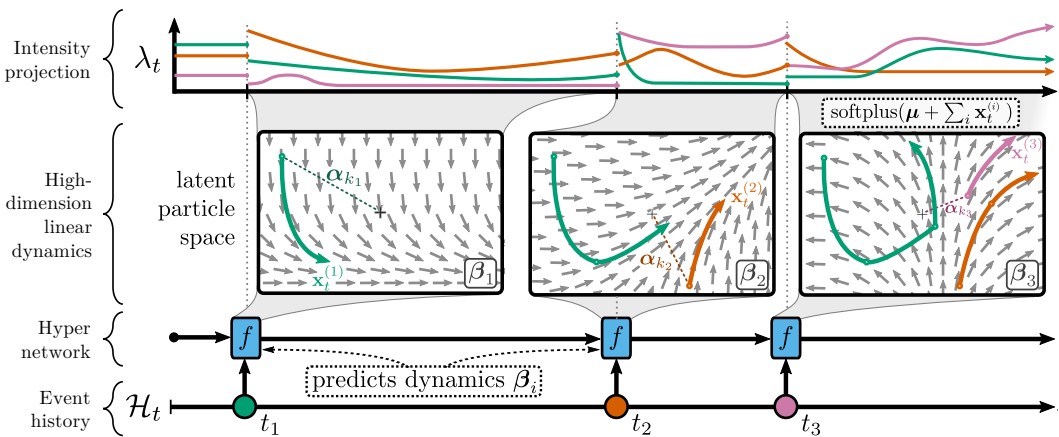

Figure 1: [Added] High-level schematic of the proposed *hyper Hawkes process* (HHP). The bottom row shows the sequence of marked events. These are input into a hypernetwork, with transition function $f$. We use a regular GRU throughout this work for the hypernetwork. The hypernetwork outputs the dynamics of a generalized linear Hawkes process, the flow fields for which are shown. These dynamics are then projected into intensity space. Crucially, these intensities can be decomposed into individual *particles* each attributed to a single event, shown as different colored lines in the flow field, which are summed over and transformed to create the intensities. At a given point in time, these particles are aggregated to produce the model's predicted intensity. This allows us to create a highly expressive model (through the time-dependence of the hypernetwork and the high-dimensional diagonalized linear dynamics), but where event-level attribution is possible through the particle-level decomposition.

**Paper Outline.**   In Section 2 we introduce the background strictly necessary for defining and understanding our hyper Hawkes process as a "black-box" MTPP model. In Section 3 we introduce the HHP as a black-box MTPP model, and evaluate its predictive performance on real-world datasets in Section 5. In Section 6 we explore and demonstrate on synthetic data how the HHP design naturally provides interpretability. We conclude in Section 7 with a critical discussion and future directions.

## 2   TEMPORAL POINT PROCESSES BACKGROUND

### 2.1   MARKED TEMPORAL POINT PROCESSES

In this paper we consider a *marked temporal point process* as a discrete and finite sequence of time-mark pairs, $\mathcal{H}_t := \{(t_i, k_i) \mid t_i \leq t \text{ for } i \in \mathbb{N}\}$, where $t_i \in \mathbb{R}$, $t_{i-1} < t_i \forall i$, $k_i \in \mathcal{M}$,

$\mathcal{M} := \{1, \ldots, K\}$, and $\mathcal{H}_t$ is referred to as an *event history*. We also define $\mathcal{H}_{t-}$ similarly to $\mathcal{H}_t$, except excluding events at *exactly* time $t$. Although we do not consider it here, note that $\mathcal{M}$ can be a more general space, such as countable or continuous.

One method for parameterizing an MTPP is using a *marked intensity process*. The intensity $\boldsymbol{\lambda}_t := [\lambda_t^1, \ldots, \lambda_t^K]^\top \in \mathbb{R}_{\geq 0}^K$ defines the rate of occurrence of events:

$$\lambda_t^k \mathrm{d}t := \ \mathbb{E} \left[ \text{event of type } k \text{ occurs in } [t, t + \mathrm{d}t] \mid \mathcal{H}_{t-} \right]. \tag{1}$$

The *total intensity* is then the rate that *any* event occurs, $\lambda_t := \sum_{k=1}^K \lambda_t^k$. It can then be shown that the log-likelihood for a sequence $\mathcal{H}_T$ is defined as (Daley and Vere-Jones, 2003, ch. 7.3):

$$\mathcal{L}(\mathcal{H}_T) := \sum_{i=1}^{N_T} \log \lambda_{t_i}^{k_i} - \int_0^T \lambda_s \mathrm{d}s. \tag{2}$$

To train an MTPP, we can optimize $\mathcal{L}(\mathcal{H}_T)$ over observed data (see, e.g., Mei and Eisner (2017)).

## 2.2 HAWKES PROCESSES

The *Hawkes process* (Hawkes, 1971) is a widely studied temporal point process that allows for the occurrence of events to encourage further occurrences soon thereafter, a property often referred to as *self-excitation*. This family of processes is characterized by an intensity that takes the form:

$$\lambda_t = \sigma \left( \int_0^{t-} \phi(t - s) \mathrm{d}N_s \right) \equiv \sigma \left( \sum_{i=1}^{N_{t-}} \phi(t - t_i) \right), \tag{3}$$

where $\sigma : \mathbb{R} \to \mathbb{R}_+$ ensures the intensity is non-negative, and the excitation function $\phi : \mathbb{R}_+ \to \mathbb{R}$ encodes the influence prior events have on the rate of occurrence for future events.

The *linear Hawkes process* is a common and appealing variant, employing an exponential kernel for the excitation function yielding: $\lambda_t = \mu + \sum_{i=1}^{N_{t-}} \alpha \exp(-\beta(t - t_i))$. where $\mu, \alpha, \beta \in \mathbb{R}_+$. The exponential kernel in particular makes the process Markovian, allowing us to describe the intensity function in the differential form of $\lambda_t = \mu + x_{t-}$ where $\mathrm{d}x_t = -\beta x_t + \alpha \mathrm{d}N_t$. This can easily be extended to accommodate $K$ categorical marks by generalizing the impulse and rate parameters into positive matrices $\boldsymbol{\alpha}, \boldsymbol{\beta} \in \mathbb{R}_+^{K \times K}$ leading to the following marked intensity process:

$$\mathrm{d}\mathbf{x}_t = -\boldsymbol{\beta}\mathbf{x}_{t-}\mathrm{d}t + \boldsymbol{\alpha}\mathrm{d}\mathbf{N}_t; \qquad \boldsymbol{\lambda}_t = \boldsymbol{\mu} + \mathbf{x}_{t-}, \tag{4}$$

where $\boldsymbol{\mu} \in \mathbb{R}_+^K$ is the vector of background intensities for different marks, $\mathrm{d}\mathbf{N}_t \in \{0, 1\}^K$ is $\mathbf{0}$ when no event occurs and a one-hot vector corresponding to the mark $k$ that occurs at time $t$, and $\mathbf{x}_0 = \mathbf{0}$. From this form, we can compute the left limit of the intensity at the next event, given a right limit $\mathbf{x}_{t_i}$ for time $t_i$, in closed form as: $\mathbf{x}_{t_{i+1}} = e^{-\boldsymbol{\beta}(t_{i+1}-t_i)}\mathbf{x}_{t_i}$, where $t_i < t_{i+1}$ and no event occurred between $t_i$ and $t_{i+1}$. Hawkes processes such as this are widely used in statistical inference settings, due to $\boldsymbol{\beta}$ and $\boldsymbol{\alpha}$ being directly interpretable by describing marginal event-to-event effects.

# 3 HYPER HAWKES PROCESSES

We now introduce the core of our hyper Hawkes process (HHP). First, we start by identifying the key mechanism that allows classical Hawkes processes to identify the effect that a given event has on future intensities. We then extend the Hawkes process to be more flexible and performant while retaining the identifiable event-level effects. Lastly, we briefly discuss how the HHP is implemented in practice. Note that we defer discussion on exactly *how* to interpret the model to Section 6.

## 3.1 EVENT-LEVEL EFFECTS IN HAWKES PROCESSES

As shown in Equation (3), classical Hawkes processes are formally defined by their rectification function $\sigma$ and their excitation function $\phi$. While $\phi$ is often parameterized to isolate marginal direct effects of events of one type to another (e.g., through $\boldsymbol{\beta}$ and $\boldsymbol{\alpha}$ for linear Hawkes processes), they actually also provide event-level attribution. In particular, $\phi(t - t_i)$ directly encodes how much the $i^{\text{th}}$ event is exciting or inhibiting an event occurrence at time $t$. $\phi(t - t_i)$ could even be generalized

to conditional $\phi_i(t; \mathcal{H}_t)$ and this effect would still be identifiable so long as the final intensity is a monotonic transform of a linear combination of these effects: $\lambda_t = \sigma(\sum_{i=1}^{N_{t-}} \phi_i(t; \mathcal{H}_{t-}))$. It should be noted that the linear Hawkes process maintains this property through the linearity of the recurrence relation, as for $t \in (t_i, t_{i+1}]$ it holds that:

$$\mathbf{x}_{t-} = e^{-\boldsymbol{\beta}(t-t_i)}\mathbf{x}_{t_i} = e^{-\boldsymbol{\beta}(t-t_i)}(\mathbf{x}_{t_i-} + \boldsymbol{\alpha}_{k_i}) = \cdots = \sum_{j=1}^{i} e^{-\boldsymbol{\beta}(t-t_j)}\boldsymbol{\alpha}_{k_j}.$$

## 3.2 Generalizing Hawkes Processes

While the linear Hawkes process (LHP) is widely used in statistical inference because of its interpretability, the model itself is not very expressive: the parameterization only allows excitation, the recurrence is directly coupled to the intensity, and the dynamics are time-invariant. We generalize the LHP by tackling these weaknesses in-turn below; all while ensuring that interpretable event-level effects are maintained by ensuring the recurrence relation remains linear.

**Increasing the Latent Dimension.** The LHP has limited expressivity because each recurrent dimension ties one-to-one with an output intensity, i.e., changing dimension $k$ in $\mathbf{x}_t$ only affects $\lambda_t^k$. This is akin to using an RNN's hidden state directly as output ($\hat{\mathbf{y}}_i := \mathbf{h}_i$) rather than applying a transformation ($\hat{\mathbf{y}}_i := \sigma(\mathbf{W}\mathbf{h}_i + \mathbf{b})$). Additionally, $\boldsymbol{\beta}$ and $\boldsymbol{\alpha}$ are parameterized to be strictly positive, thus only allowing for excitation and not inhibition. We address both of these issues this by lifting the dynamics into a latent space:

$$\mathrm{d}\mathbf{x}_t = -\boldsymbol{\beta}\mathbf{x}_{t-}\mathrm{d}t + \boldsymbol{\alpha}\mathrm{d}\mathbf{N}_t; \qquad \boldsymbol{\lambda}_t = \sigma(\boldsymbol{\mu} + \mathbf{W}\mathbf{x}_{t-}), \tag{5}$$

where $\boldsymbol{\beta}, \boldsymbol{\alpha} \in \mathbb{R}^{d \times d}$, $\mathbf{W} \in \mathbb{R}^{d \times K}$, $\boldsymbol{\mu} \in \mathbb{R}^K$, and $\sigma(z) = \log(1 + e^z)$ is the softplus. This decouples the recurrent dimension $d$ from the number of marks $K$, while $\sigma$ ensures inhibitory effects still result in a nonnegative intensity. This increases expressivity when $d > K$, or, prevents $\mathcal{O}(K^2)$ parameter scaling when $K$ is large by setting $d < K$. While $\boldsymbol{\alpha}$ and $\boldsymbol{\beta}$ become less directly interpretable, event-specific effects remain identifiable through the projection of the latent particles $\mathbf{W} \exp(-\boldsymbol{\beta}(t - t_i))\boldsymbol{\alpha}_{k_i}$.

**Adding Time-Variation through Hypernetworks.** Even in this expanded space, the model is limited in its expressivity because of the linearity of the recurrence relation and the time-invariant dynamics. In general, RNNs (such as GRUs) do not suffer from this problem due to the nonlinearity applied when propagating the hidden state. Unfortunately, this is undesirable as it removes identifiable contributions, complicating the model interpretability (see the extensive literature on interpreting non-linear RNNs in even simple tasks, e.g. Miconi (2017); Maheswaranathan and Sussillo (2020)).

An alternative, therefore, is to retain the linearity of the recurrence, but make the *dynamics* vary across time, i.e., $\boldsymbol{\beta}$ becomes $\boldsymbol{\beta}_t$. We could trivially make the dynamics mark-specific, only depending on the most recent event, but subsequent events will critically suppress the enduring influence that each previous event can exert. Therefore we instead introduce a *hypernetwork* (Ha et al., 2017), $f_\theta$, that predicts the dynamics as a function of the history. The resulting recurrence relation is:

$$\mathrm{d}\mathbf{x}_t = -\boldsymbol{\beta}_t\mathbf{x}_{t-}\mathrm{d}t + \boldsymbol{\alpha}\mathrm{d}\mathbf{N}_t; \qquad \boldsymbol{\beta}_t = f_\theta(\mathcal{H}_t); \qquad \boldsymbol{\lambda}_t = \sigma(\boldsymbol{\mu} + \mathbf{W}\mathbf{x}_{t-}). \tag{6}$$

We experimented with predicting history-dependent $\boldsymbol{\alpha}$, however, we found it only had a marginal impact on performance, and greatly detracts the interpretability arguments presented in Section 6. We therefore do not create history-dependent impulses, and only utilize fixed impulses. Even with static impulses, Equation (6) now defines highly expressive non-linear latent dynamics as a function of individual event sequences. We use a standard multi-layer GRU as the hypernetwork throughout. [Added] This is because the GRUs widely-available and highly optimized implementation, linear work complexity, and good generalized performance. The "HHP" architecture is not fundamentally tied to this choice, however, and exploration of alternative hypernetwork architectures is interesting future work.

**Efficient and Expressive Parameterization.** The final component is selecting a parameterization for how $f_\theta$ generates $\boldsymbol{\beta}_t$ that is both efficient and expressive. For efficiency, we use closed-form updates similar to the time-invariant setting by making $\boldsymbol{\beta}_t$ piecewise-constant between events. To avoid an expensive matrix exponential, we use a diagonal parameterization of $\boldsymbol{\beta}$. This leads to $\boldsymbol{\beta}_t := \mathbf{V}_i\mathbf{D}_i\mathbf{V}_i^{-1}$ for $t \in (t_i, t_{i+1}]$, where $\mathbf{V}_i, \mathbf{D}_i \in \mathbb{C}^{d \times d}$ and $\mathbf{D}_i$ is diagonal, representing the

eigenvectors and eigenvalues of $\boldsymbol{\beta}_t$ respectively. The parameters $\mathbf{V}_i$ and $\mathbf{D}_i$ are predicted by the hypernetwork $f_\theta(\mathcal{H}_{t_i})$. For stability, we parameterize $\Re(\mathbf{D}_i) < 0$ and $\mathbf{V}_i$ to be unitary (Jing et al., 2017), so that $\mathbf{V}_i^{-1} \equiv \mathbf{V}_i^*$. This leads to the final HHP update equation:

$$\mathbf{x}_t = \mathbf{V}_i e^{\mathbf{D}_i(t-t_i)} \mathbf{V}_i^* \mathbf{x}_{t_i} + \boldsymbol{\alpha}_{k_{i+1}} \mathbb{1}(t = t_{i+1}); \quad \mathbf{V}_i, \mathbf{D}_i = f_\theta(\mathcal{H}_t); \quad \boldsymbol{\lambda}_t = \sigma(\boldsymbol{\mu} + \mathbf{W}\mathbf{x}_{t-}), \quad (7)$$

for $t \in (t_i, t_{i+1}]$, and $e^{\mathbf{D}_i(t-t_i)}$ is applied element-wise as $\mathbf{D}_i$ is a diagonal matrix. Extensive details on this and the implementation of the architecture can be found in Appendix A.

While keeping $\mathbf{V}_i$ constant would be simpler and more computationally efficient, we instead update the eigenvectors after each event to enhance expressiveness. Because the Hawkes process is a state-space model (Chang et al., 2025), the results of Merrill et al. (2024a) apply: if the time-varying dynamics $\boldsymbol{\beta}_i$ are not simultaneously diagonalizable (i.e., $\mathbf{V}_i \neq \mathbf{V}_j$), then despite the linearity of the recurrence, the model exhibits state-tracking capabilities comparable to those of RNNs.

**Summary.** We briefly summarize the HHP model we have proposed. A (nonlinear) hypernetwork consumes the event history, and outputs piecewise constant dynamics parameters for a high-dimensional linear recurrence with learned impulses at events. We parameterize the dynamics in a per-event eigenbasis also predicted by the hypernetwork. This allows for efficient closed-form computation of updates to the latent state in continuous time to time points of interest. We then decode the latent state by projecting it into the intensity space and applying a rectification function to ensure intensities are non-negative. This allows us to define a highly flexible intensity-based neural MTPP that we can efficiently evaluate at any time point, but that has a (conditionally) linear recurrence which will serve as a "bottleneck" that we can inspect, as we explore in Section 6.

# 4 RELATED WORKS

**Neural MTPPs.** Marked temporal point processes (MTPPs) model both event timing and type in continuous time, often via intensity functions (Daley and Vere-Jones, 2003). Early work relied on parametric forms, such as self-exciting Hawkes processes (Hawkes, 1971; Liniger, 2009). Recent advances leverage neural architectures for flexible conditional intensity modeling, including RNN-based models (Du et al., 2016; Mei and Eisner, 2017), CNNs (Zhuzhel et al., 2023), transformer-based approaches (Zhang et al., 2020; Zuo et al., 2020; Yang et al., 2022), and deep state space models (Gao et al., 2024; Chang et al., 2025). Intensity-free alternatives have also emerged, using normalizing flows (Shchur et al., 2020; Zagatti et al., 2024), neural processes (Bae et al., 2023), and diffusion-based models (Zeng et al., 2023). Despite these developments, intensity-based methods remain dominant due to their structural simplicity and fewer modeling assumptions.

**Interpretable MTPPs.** The original Hawkes process offered a transparent parameterization (Hawkes, 1971), but many neural MTPPs (e.g., transformer Hawkes (Zuo et al., 2020), intensity-free TPP (Shchur et al., 2020)) prioritize predictive accuracy over interpretability. Meng et al. (2024) introduce a single-layer attention model, enabling easy inspection of pairwise contributions, while Song et al. (2024) propose neural ODEs parameterized by event type which are aggregated post-activation. These choices aid interpretability but restricts interactions to pairwise and excitatory interactions. Our HHP addresses these limitations by supporting both excitatory and inhibitory effects, while still capturing higher-order interactions among multiple events through the hypernetwork. Rule-based approaches (Li et al., 2022; Yang et al., 2024; Li et al., 2020) provide interpretable boolean rules, but require large rule sets or soft weighting, which can reduce clarity and expressivity.

**Deep State Space Models and State Tracking.** Chang et al. (2025) identified a connection between conventional linear Hawkes processes and modern deep state space models (Gu et al., 2022; Smith et al., 2022; Gu and Dao, 2023). Their S2P2 architecture uses deep stacks of linear-Hawkes-like layers, with only the time-constants of the dynamics matrix being data-dependent. In contrast, we use a single linear Hawkes layer with the both time constants $\mathbf{D}$ *and* eigenvectors $\mathbf{V}$ being data dependent. This was inspired by a finding by Merrill et al. (2024b) that found that *non-simultaneously diagonalizable* dynamics (i.e., having variable eigenvectors) greatly increased SSM expressivity.

## 5 EXPERIMENTS

We evaluate the HHP on common TPP benchmarks, finding that our approach achieves state-of-the-art predictive performance, even before consideration of interpretability. Please see Appendix B for full hyperparameter selections and search configurations for all models and experiments.

**Datasets.** We evaluate our HHP and baseline models on seven widely used, real-world MTPP datasets. These datasets are: Amazon reviews (Ni et al., 2019), Retweet cascades (Zhao et al., 2015), Taxi pickups (Whong, 2014), Taobao purchases (Xue et al., 2022), StackOverflow posts (Leskovec and Krevl, 2014), Last.fm listening patterns (Celma Herrada et al., 2009), MIMIC-II medical events (Saeed et al., 2002). We provide more details on each dataset and their preparation in Appendix B.

**Evaluation Metrics.** We evaluate the per-event log-likelihood as our primary measure of performance. This is both what the models are trained to optimize and is a proper scoring metric (Heinrich-Mertsching et al., 2024). In Table 5 we separate the log-likelihood into the likelihoods for both times and marks to further interrogate the models performance. As more interpretable summary metrics, we also compute the RMSE of the next event time prediction and the average accuracy of the next mark type prediction. Finally, we also evaluate *calibration*, which provides a measure of how well the model quantifies the uncertainty in its predictions (Bosser and Taieb, 2023). We defer the calibration results to Appendix B. In Table 1, following (Chang et al., 2025), we also provide a "composite metric", aggregating performance across all metrics on all datasets.

**Ablations.** As mentioned previously, the HHP generalizes the linear Hawkes process in various ways. To assess each extension, we also measure the performance of three different ablations: (i) HHP$_{\neg\text{Stateful}}$, which disables the "statefulness" by setting the eigenvectors to a learned constant basis[1], $\mathbf{V}_i = \mathbf{V}$; (ii) HHP$_{\neg\text{Hyper}}$, which disables the hypernetwork entirely, $\boldsymbol{\beta}_i = \boldsymbol{\beta}$; and (iii) HHP$_{\neg\text{Latent}}$, which both disables the hypernetwork *and* removes the latent space, setting $d = K$ and $\mathbf{W} = \mathbf{I}$.

**Results.** Results are presented in Table 2. Most importantly, we see that our HHP performs on par or better than almost all existing baseline models across all datasets, only being narrowly outperformed on average by S2P2 (Chang et al., 2025) in terms of log-likelihood. Notably, HHP achieves this level of performance while using, on average, 54% fewer parameters than S2P2 across datasets. For both next event time and mark prediction, HHP is the clear leading model with average ranks of 1.4 and 1.7, respectively. [Added] We defer full calibration results to the appendix, but we find that all models are comparably calibrated, with no stand-out winner. Furthermore, a better calibrated model is not a guarantee of better predictions, and therefore should always be considered alongside purely predictive metrics.

Interestingly, we see that while the statefulness of the HHP does have a marked impact on performance, even without it, the model HHP$_{\neg\text{Stateful}}$ is still competitive. Perhaps even more surprising is that, even with static dynamics, HHP$_{\neg\text{Hyper}}$ outperforms several baselines. This suggests that a major bottleneck in the conventional Hawkes process was the tying of latent dimensions to the mark-space, as well as that the basic form of the Hawkes process provides a strong inductive bias for MTPPs.

## 6 ON INTERPRETABILITY

As discussed in Sections 2 and 3, a key feature of both the linear Hawkes process and our proposed HHP is the linear recurrence structure. This structure enables us to view the model equivalently as a recurrence; or through a particle or branching process perspective where each event contributes a distinct, trackable influence on future predictions. By leveraging this property, we can attribute model outputs to specific past events, providing a key foundation for interpretability. In the following, we introduce practical tools that exploit this structure and demonstrate their utility on synthetic data.

### 6.1 PRACTICAL INTERPRETABILITY TOOLS FOR HHP

Using the linear recurrence of HHP, we can directly probe how individual events influence the model's predictions. In this subsection, we introduce a suite of practical tools that leverage this structure,

---

[1][Added] Note: this is also equivalent to the case where $\beta_t$ is constrained to be a diagonal matrix, as $B$ and $C$ are unconstrained, effectively setting $V_i = \mathbb{I}$.

Table 2: Quantitative results for TPP models across datasets. **Bold** entries correspond to best result, and underlined for second-best, amongst baselines and main proposed method*. [Added] Shown are the means and (standard deviations) across five randomly initialized models.

(a) Per event log-likelihood. Higher log-likelihood values indicate better performance.

| Model | Amazon | Retweet | Taxi | Taobao | StackOverflow | Last.fm | MIMIC-II | Average Rank* ($\downarrow$) |
|---|---|---|---|---|---|---|---|---|
| | | | | | Per Event Log-Likelihood, $\mathcal{L}_{\text{Total}}$ (nats) ($\uparrow$) | | | |
| RMTPP (Du et al., 2016) | -2.136 (0.003) | -7.098 (0.217) | 0.346 (0.002) | 1.003 (0.004) | -2.480 (0.019) | -1.780 (0.005) | -0.472 (0.026) | 7.3 |
| NHP (Mei and Eisner, 2017) | 0.129 (0.012) | **-6.348** (0.000) | 0.514 (0.004) | 1.157 (0.004) | -2.241 (0.002) | -0.574 (0.011) | 0.060 (0.017) | 4.0 |
| SAHP (Zhang et al., 2020) | -2.074 (0.029) | -6.708 (0.029) | 0.298 (0.057) | 1.168 (0.028) | -2.341 (0.058) | -1.646 (0.083) | -0.677 (0.072) | 6.6 |
| THP (Zuo et al., 2020) | -2.096 (0.002) | -6.659 (0.007) | 0.372 (0.002) | 0.790 (0.002) | -2.338 (0.014) | -1.712 (0.011) | -0.577 (0.011) | 6.6 |
| IFTPP (Shchur et al., 2020) | 0.496 (0.002) | -10.344 (0.016) | 0.453 (0.002) | **1.318** (0.017) | -2.233 (0.009) | **-0.492** (0.017) | 0.317 (0.052) | 3.6 |
| AttNHP (Yang et al., 2022) | 0.484 (0.077) | -6.499 (0.028) | 0.493 (0.009) | 1.259 (0.022) | -2.194 (0.016) | -0.592 (0.051) | -0.170 (0.077) | 3.9 |
| S2P2 (Chang et al., 2025) | **0.781** (0.011) | -6.365 (0.003) | **0.522** (0.004) | 1.304 (0.039) | **-2.163** (0.009) | -0.557 (0.046) | 0.919 (0.069) | **1.7** |
| **HHP** (Ours) | 0.616 (0.016) | -6.366 (0.003) | 0.520 (0.003) | 1.232 (0.014) | -2.209 (0.006) | -0.515 (0.006) | **1.314** (0.048) | 2.4 |
| HHP$_{\neg\text{Stateful}}$ | 0.606 (0.006) | -6.370 (—) | 0.508 (0.004) | 1.249 (0.004) | -2.195 (0.006) | -0.572 (—) | 1.114 (0.032) | 2.7 |
| HHP$_{\neg\text{Hyper}}$ (Ablations) | 0.514 (0.012) | -6.796 (—) | 0.469 (0.001) | 1.224 (0.002) | -2.246 (0.004) | -1.028 (—) | 0.305 (0.036) | 4.1 |
| HHP$_{\neg\text{Latent}}$ | -0.170 (0.061) | -6.880 (—) | 0.237 (0.024) | 1.150 (0.005) | -2.374 (0.002) | -1.390 (—) | -0.533 (0.010) | 6.1 |

(b) Prediction RMSE of the next event time prediction. Lower RMSE values indicate better performance.

| Model | Amazon | Retweet | Taxi | Taobao | StackOverflow | Last.fm | MIMIC-II | Average Rank* ($\downarrow$) |
|---|---|---|---|---|---|---|---|---|
| | | | | | RMSE, $\mathcal{L}_{\text{Total}}$ ($\downarrow$) | | | |
| RMTPP (Du et al., 2016) | 0.338 (0.000) | 16488 (070.5) | 0.283 (0.001) | 0.126 (0.000) | 1.049 (0.003) | 15.873 (0.000) | 0.749 (0.010) | 5.1 |
| NHP (Mei and Eisner, 2017) | 0.339 (0.000) | 15911 (004.0) | 0.282 (0.001) | 0.126 (0.000) | 1.019 (0.001) | 15.733 (0.008) | 0.726 (0.001) | 3.1 |
| SAHP (Zhang et al., 2020) | 0.335 (0.001) | 16102 (062.4) | 0.290 (0.008) | 0.126 (0.000) | 1.031 (0.011) | 15.757 (0.007) | 1.142 (0.198) | 4.9 |
| THP (Zuo et al., 2020) | 0.332 (0.000) | 16268 (018.7) | 0.285 (0.001) | **0.125** (0.000) | 1.033 (0.005) | 15.871 (0.000) | 0.768 (0.005) | 4.6 |
| IFTPP (Shchur et al., 2020) | 0.327 (0.000) | 16625 (000.2) | 0.362 (0.178) | **0.125** (0.000) | 1.340 (0.724) | 16.508 (0.555) | 0.767 (0.029) | 5.4 |
| AttNHP (Yang et al., 2022) | 2.656 (1.950) | 16171 (284.2) | 1.739 (0.422) | 0.130 (0.000) | 1.256 (0.030) | 15.865 (0.017) | 0.860 (0.022) | 6.7 |
| S2P2 (Chang et al., 2025) | 0.327 (0.000) | 15987 (013.7) | **0.281** (0.000) | 0.126 (0.000) | **1.014** (0.001) | **15.720** (0.000) | 0.894 (0.054) | 2.6 |
| **HHP** (Ours) | **0.324** (0.000) | **15590** (011.3) | **0.281** (0.001) | 0.127 (0.001) | 1.016 (0.001) | 15.741 (0.033) | **0.714** (0.013) | 2.3 |
| HHP$_{\neg\text{Stateful}}$ | 0.325 (0.000) | 15559 (—) | 0.283 (0.000) | 0.125 (0.000) | 1.017 (0.002) | 15.793 (—) | 0.720 (0.014) | 1.9 |
| HHP$_{\neg\text{Hyper}}$ (Ablations) | 0.328 (0.001) | 15516 (—) | 0.283 (0.000) | 0.125 (0.000) | 1.025 (0.000) | 15.831 (—) | 0.772 (0.006) | 2.9 |
| HHP$_{\neg\text{Latent}}$ | 0.339 (0.002) | 15672 (—) | 0.294 (0.002) | 0.126 (0.000) | 1.038 (0.001) | 15.888 (—) | 0.804 (0.010) | 4.7 |

(c) Mark prediction accuracy for the next event. Higher accuracy values indicate better performance.

| Model | Amazon | Retweet | Taxi | Taobao | StackOverflow | Last.fm | MIMIC-II | Average Rank* ($\downarrow$) |
|---|---|---|---|---|---|---|---|---|
| | | | | | Accuracy, $\mathcal{L}_{\text{Total}}$ ($\uparrow$) | | | |
| RMTPP (Du et al., 2016) | 30.8 (0.1) | 53.4 (0.6) | 91.4 (0.1) | 60.9 (0.1) | 45.6 (0.3) | 52.5 (0.1) | 92.3 (0.3) | 6.6 |
| NHP (Mei and Eisner, 2017) | 39.4 (0.1) | **61.4** (0.0) | 92.9 (0.1) | **61.5** (0.2) | 47.1 (0.1) | 56.5 (0.1) | 94.3 (0.0) | 2.3 |
| SAHP (Zhang et al., 2020) | 32.4 (1.0) | 57.5 (2.2) | 91.4 (0.7) | 60.5 (0.2) | 44.7 (2.0) | 51.8 (0.7) | 86.8 (0.9) | 7.1 |
| THP (Zuo et al., 2020) | 34.6 (0.1) | 60.2 (0.1) | 91.8 (0.0) | 60.0 (0.0) | 46.6 (0.2) | 53.3 (0.1) | 90.9 (0.2) | 6.1 |
| IFTPP (Shchur et al., 2020) | 35.9 (0.1) | 50.4 (2.5) | 91.8 (0.0) | 61.0 (0.1) | 45.6 (0.1) | 56.4 (0.1) | 93.4 (0.1) | 5.1 |
| AttNHP (Yang et al., 2022) | 38.9 (0.9) | 60.7 (0.2) | 92.6 (0.1) | 61.3 (0.2) | **48.2** (0.2) | 55.8 (0.6) | 92.9 (0.6) | 3.6 |
| S2P2 (Chang et al., 2025) | 40.7 (0.0) | 61.3 (0.0) | **93.1** (0.1) | 61.1 (0.1) | 47.5 (0.3) | 55.8 (0.4) | 96.0 (0.4) | 2.4 |
| **HHP** (Ours) | **40.8** (0.1) | 61.2 (0.0) | 93.0 (0.0) | 61.4 (0.1) | 47.1 (0.1) | **56.6** (0.0) | **96.9** (0.2) | **1.9** |
| HHP$_{\neg\text{Stateful}}$ | 40.9 (0.1) | 61.1 (—) | 92.9 (0.1) | 61.7 (0.0) | 47.3 (0.0) | 56.4 (—) | 96.8 (0.5) | 1.9 |
| HHP$_{\neg\text{Hyper}}$ (Ablations) | 40.3 (0.1) | 57.2 (—) | 92.4 (0.0) | 61.4 (0.0) | 46.8 (0.1) | 53.5 (—) | 95.0 (0.2) | 3.6 |
| HHP$_{\neg\text{Latent}}$ | 34.0 (1.5) | 57.6 (—) | 91.2 (0.2) | 60.6 (0.1) | 46.6 (0.0) | 54.3 (—) | 90.5 (0.3) | 5.9 |

* Ablations are not included in main rankings. Ranks for ablations compare solely that ablations performance relative to the baselines.

enabling us to quantify, visualize, and interpret the contributions of specific events or groups of events to the predicted intensity. These tools provide actionable insight into the model's internal mechanism, going beyond basic aggregate parameter inspection to per-event-level attribution instead.

**Particle View.** A central feature of both the linear Hawkes process (LHP) and the HHP is that the model's latent state at any time can be decomposed into a sum of event-specific contributions, which we refer to as *particles*. Each particle encodes how the influence of a single past event on the current state and predicted intensity evolves over time.

In the LHP, the effect of the $i^{\text{th}}$ event at time $t$ is $e^{-\boldsymbol{\beta}(t-t_i)}\boldsymbol{\alpha}_{k_i}$, which we will denote as $\mathbf{x}_t^{(i)}$, and the overall intensity is $\boldsymbol{\lambda}_t = \boldsymbol{\mu} + \sum_{i=1}^{N_{t-}} \mathbf{x}_t^{(i)}$. Each dimension of a particle encodes the degree to which that event excites or inhibits future occurrences of a specific mark.

Our HHP preserves this structure, but with more expressive, history-dependent dynamics:

$$\mathbf{x}_t^{(i)} := \mathbf{W}\left(\prod_{j=i}^{N_t} \mathbf{V}_j e^{\mathbf{D}_j(\min\{t,t_{j+1}\}-t_j)}\mathbf{V}_j^*\right)\boldsymbol{\alpha}_{k_i}; \quad \boldsymbol{\lambda}_t \equiv \sigma(\boldsymbol{\mu} + \sum_{i=1}^{N_{t-}}\mathbf{x}_t^{(i)}) \tag{8}$$

where the product is taken from right to left in chronological order from event $i$ to event $N_t$. This captures how each event's initial impact evolves through subsequent adaptive transformations. All $\mathbf{D}$ and $\mathbf{V}$ values can be computed for the sequence, and then all particle positions can be efficiently

computed in parallel.[2] This decomposition allows us to isolate the contribution of each event to the model's latent state and, consequently, to the predicted intensity—providing direct insight into how the model encodes memory, excitation, and inhibition across the event sequence.

*Reflection:* A key aspect of HHP's design is that, after each new event, the updated dynamics apply uniformly to all existing particles. Because there is no skip connection from the hypernetwork to the output, the model cannot bypass the aggregation of particles to directly predict intensities; instead, it must learn meaningful, event-driven dynamics that govern excitation and inhibition. As a result, the hypernetwork orchestrates the implicit evolution and decay of particles, which can be viewed a form of working memory, maintaining relevant information and enabling flexible prediction.

**Leave-one-out.** While particles isolate the effects that an event has on predictions through the model, the values they hold are inherently *contextual* since they never act upon the outputs in isolation. Due to the nonlinear rectification $\sigma$, the influence of a particle $\mathrm{d}\lambda_t/\mathrm{d}\mathbf{x}_t^{(i)}$ depends on the superposition of all other particles. To account for this, taking inspiration from the diagnostic tools DFBETA and DFFIT used in linear regression (Belsley et al., 2005), we introduce leave-one-out estimators of a particle's influence on the output intensity termed DF$\boldsymbol{\lambda}$ where:

$$\mathrm{DF}\boldsymbol{\lambda}_t^{(i)} := \boldsymbol{\lambda}_t - \sigma\left(\boldsymbol{\mu} + \sum_{j=1}^{N_t}\mathbf{x}_t^{(j)}\mathbb{1}(j \neq i)\right). \tag{9}$$

Here, DF$\boldsymbol{\lambda}_t^{(i)}$ represents how the model chose to utilize the $i^{\text{th}}$ event's particle to change the output intensity. Values of 0 indicate no instantaneous influence on the output, positive values indicate excitement, and negative values indicate inhibition. Note that we use parentheses to represent event indices, not to be confused with mark-specific values. We can also compute a "total intensity" version, $\sum_{m=1}^{M}\mathrm{DF}\boldsymbol{\lambda}_t^{(i),m}$, corresponding to the amount of influence any event has on the occurrence of an event of any type in the future. We visualize this quantity in Figure 2

**Cumulative Effects.** While DF$\boldsymbol{\lambda}$ captures instantaneous influence, it is often useful to understand the total effect an event has over time. By integrating the influence of a particle across the prediction horizon, we can capture its cumulative impact on the expected number or timing of future events. We denote this as DF$\boldsymbol{\Lambda}$ where $\mathrm{DF}\boldsymbol{\Lambda}_t^{(i)} := \int_0^t \mathrm{DF}\boldsymbol{\lambda}_s^{(i)}\mathrm{d}s$. It should be noted that $\boldsymbol{\Lambda}_t := \int_0^t \boldsymbol{\lambda}_s\mathrm{d}s$ is equivalent to $\mathbb{E}[N_t]$, thus we can conclude that DF$\boldsymbol{\Lambda}$ exists on the same scale as number of events. Furthermore, it can be thought of as how many events the particle encouraged or inhibited, in expectation, when acted upon by the model. Likewise, integrating $\mathrm{DF}|\boldsymbol{\lambda}_t^{(i)}|$ can measure the total cumulative influence of a given particle, regardless if it excites or inhibits.

**Group Influences.** Finally, the linear structure of HHP enables us to extend these analyses to groups of events. By jointly removing or modifying sets of particles, we can attribute model predictions to combinations of events—such as all events of a certain type or within a specific time window—shedding light on higher-order interactions and collective effects. This is achieved simply by removing multiple particles when calculating the above metrics, e.g., $\mathrm{DF}\boldsymbol{\lambda}_t^{(A)} := \boldsymbol{\lambda}_t - \sigma\left(\boldsymbol{\mu} + \sum_{j=1}^{N_{t^-}}\mathbf{x}_{t^-}^{(j)}\mathbb{1}(j \notin A)\right)$ for $A \subset \mathbb{N}$.

These tools collectively enable model-level event-attribution, providing understanding of how previous events influence future predicted intensities in models with rich dynamics. Such analysis can assist in describing various temporal patterns that the model relies on in its predictions, which is useful for providing descriptions of *how* models generate predictions. This addresses the gap identified earlier regarding interpretability and performance. [Added] Crucially, the number of particles required is equal to the number of observations, and the leave-one-out (or leave-n-out) estimators are linear combinations, and hence the analysis is very computationally cheap. The extensibility of this analysis is a huge opportunity, because it allows the influence and interaction of multiple different events the be efficiently analyzed, allowing, for instance, diagnosis comorbidities in healthcare. This ameliorates the combinatorial cost of performing this analysis in conventional black-box neural MTPP models.

---

[2]This particle decomposition is a conceptual and interpretive tool; during training and general inference, only the total state $\mathbf{x}_t$ is maintained, so there is no computational overhead from tracking individual particles.

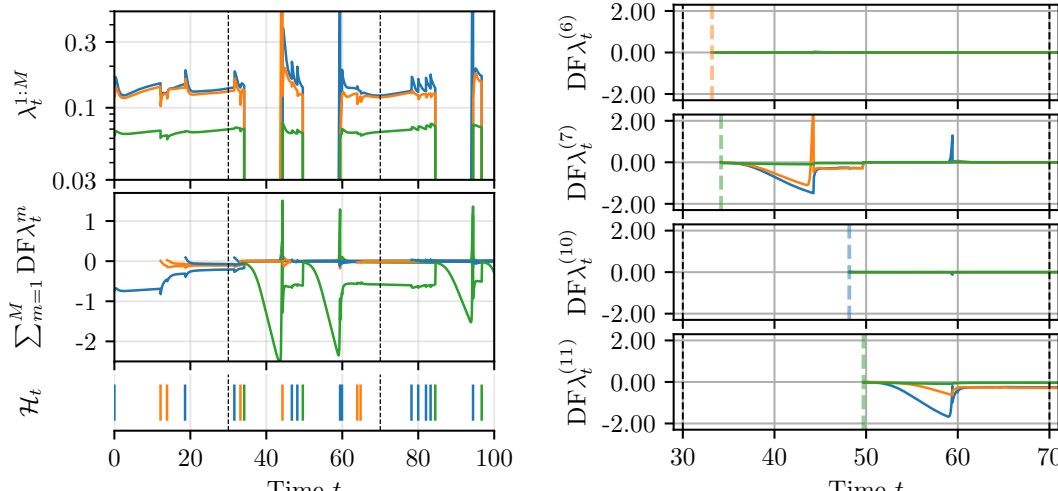

Figure 2: Visualizations of interpretability results presented in Section 6.1. Bottom left shows a sequence of events where a blue or orange mark is repeated after a predictable time after a green mark occurs. The top left is the model's predicted marked intensities. Middle left showcases the *total* DF$\lambda$ values per event, with lines colored by the mark that spawned the particle. Right plots show mark-specific DF$\lambda$ trajectories for four particles in the highlighted time range $(30, 70)$.

## 6.2 EMPIRICAL EXPLORATION

We now explore these estimators using a synthetic memory task, shown in Figure 2. Marks are drawn sequentially from a homogeneous Poisson process, until a green "trigger" event is drawn, which causes the previous mark to be repeated a predictable time later. In Figure 2 (top left) we show the overall intensities learned by the model. We see that it successfully captures the homogeneous Poisson occurrence of events, the zero intensity after trigger event, and the sharp spike of intensity after the delay period, before returning to normal.

We then explore a learned HHP model using the leave-one-out estimators introduced above. In the middle left panel, we show the time-evolution of the instantaneous *total* intensity attributed to each source event. The occurrence of a green trigger event (specifically and identifiably) dramatically inhibits the other marks during the delay period (seen by the negative green line during the delay period), before causing the intensity to rise at the target event (seen by the green lines sharply spiking upward), before returning to a quiescent position. We see also the blue and orange marks do not actually contribute to any other intensity, indicating that the rise in the *correct* marks intensity is attributed to the green event and the hypernetwork. This reduction highlights those events and particles that are most responsible for future events and how.

In the right-hand figure we unpack the DF$\lambda$ on a per-mark basis for examples of events that are identified as responsible and not-responsible (two trigger events, 7 and 11; two non-trigger events, 6 and 10). We see the non-trigger events have almost no influence on *any* event, and trigger events mediate the intensities of subsequent events as expected for (e.g.) the first response, driving exciting orange at the right time and inhibiting blue. However, interestingly, for the second trigger, we see that *both* trigger events are used to generate the swing in intensity for the response, highlighting that truly separating causal effects in a flexible model, without direct injection of domain knowledge or additional constraints, is not guaranteed. This is something we discuss below. Even with this, we believe these estimators offer a unique and direct way to begin to understand the mechanisms that the model uses to generate predictions, in a way that is not possible with other neural MTPPs. For more details on this exploration, as well as a full analysis of another task, please see Appendix C.

## 6.3 INTERPRETING INTERPRETABILITY

Our notion of interpretability aligns in part with mechanistic interpretability as defined by Bereska and Gavves (2024): we expose and interpret part of the actual computation used to produce outputs.

The linear recurrence actually acts as a convenient bottleneck, allowing these meaningful probes to be defined (interestingly also exploiting to linearity and superposition as defined by Bereska and Gavves (2024)). Our description is however an incomplete interpretation, as particularly the internal mechanisms in the GRU remain opaque. Another perspective, following Shmueli (2010), is that the HHP is a *predictive model*, not an *explanatory model*: it forecasts future events, but does not replicate the true generative process. Our interpretability constructs therefore *describe* the model's solution strategy, and not causal relationships. In short, HHP is a predictive model that also offers a mechanistic description of its internal computation, and should not be confused with extracting true causal relationships. To achieve causal understanding would require not just an architectural change but also a shift in the underlying learning procedure itself and the injection of domain knowledge.

## 7 CONCLUSION

In this paper, we introduced the Hyper Hawkes Process (HHP), an intensity-based MTPP model that leverages a hypernetwork to predict the dynamics of a generalized Hawkes process. This design achieves state-of-the-art predictive performance, enables efficient computation, and exposes key internal variables that offer a window into its learned computational mechanism. This is unlike most classical models, which trade performance for interpretability; and most neural MTPP models, which sacrifice interpretability for performance. Our HHP aims to combine the best of both worlds: flexibility, accuracy, and interpretable structure.

However, put simply: interpreting highly flexible neural models is challenging. Our results show that it is possible to design a model that is both expressive and more interpretable than alternatives. However, the interpretability we achieve is nuanced, and requires careful analysis to extract meaningful information. Future work will focus on building systematic methods to leverage these exposed variables for domain-specific analysis, integrating them into practical workflows, and exploring how these mechanisms can guide model design. These steps will move HHP from a highly performant model, toward a broadly useful tool for understanding complex event dynamics.

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

## TABLE OF CONTENTS

# A FULL MODEL & IMPLEMENTATION DETAILS

We now summarize the model referred to as *the* Hyper Hawkes Process (HHP) throughout this paper. Its overall architecture is illustrated in Figure 3. The HHP is a recurrent neural marked temporal point process (MTPP) model, composed of two key components: a non-linear Hawkes process, denoted by $f$, and a hypernetwork, denoted by $h_\phi$.

## A.1 RECURRENT UPDATE MECHANISM

We begin by detailing how the HHP transitions from the $i$-th event to the $(i+1)$-th event. This step is highlighted in red in Figure 3, where $i = 1$ and $i + 1 = 2$. Subscripts indicate the temporal position relative to the $i$-th event: variables with subscript $i$ refer to the right limit (i.e., immediately *after* the event), while $\mathbf{x}_{i-}$ denotes the left limit (i.e., just *before* the event).

The update begins by decoding the hypernetwork state from the previous iteration, $\mathbf{z}_i$, which emits the Hawkes parameters $\mathbf{V}_i$ and $\mathbf{D}_i$ for the current step. Importantly, the $(i+1)$-th event is not yet introduced to preserve causality.

Using these parameters, we update the latent Hawkes state using the first component of Equation (7):

$$\mathbf{x}_{i+1} = \mathbf{V}_i e^{\mathbf{D}_i(t-t_i)} \mathbf{V}_i^* \mathbf{x}_{t_i}$$

applying it to the previous right-limit of the recurrent state $\mathbf{x}_i$ to produce the left-limit of the next state $\mathbf{x}_{(i+1)-}$. This update is a function of the hypernetwork-emitted dynamics and the time interval between events $t_{i+1}$ and $t_i$. We use the right-limit of the state $\mathbf{x}_i$, which already includes the impulse from the previous event. The update is computationally efficient due to the diagonal structure of $\mathbf{D}_i$.

The updated left-limit state is then projected into the output space via:

$$\boldsymbol{\lambda}_i = \sigma(\mathbf{W}\mathbf{x}_{(i+1)-} + \mathbf{b}),$$

where $\mathbf{W} \in \mathbb{R}^{K \times d}$ and $\boldsymbol{\mu} \in \mathbb{R}^K$. We use the element-wise softplus function $\sigma(a) = \log(1 + e^a)$ to ensure non-negativity. The resulting intensity $\boldsymbol{\lambda}_i$ is used to compute the log-likelihood in Equation (2).

Next, we update the Hawkes state to its new right limit by adding the mark-specific impulse:

$$\mathbf{x}_i = \mathbf{x}_{i-} + \boldsymbol{\alpha}_{k_i}.$$

Finally, we roll forward the hypernetwork state using the current event:

$$\mathbf{z}_i = h(\mathbf{z}_{i-1}, t_{i+1} - t_i, k_i),$$

which will be used in the next iteration. We input the logarithm of the time difference and a one-hot encoding of the mark into the hypernetwork.

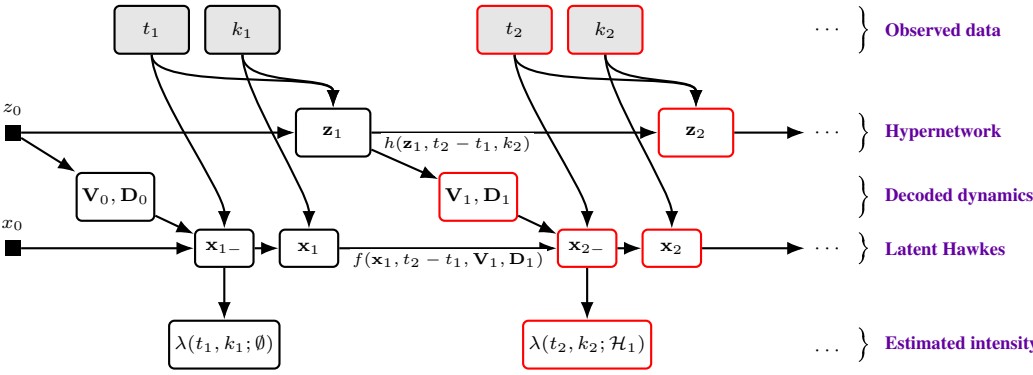

Figure 3: The full hyper Hawkes process architecture. We highlight data that is conditioned on with shaded boxes, and the variables that are updated/used in a single iteration, i.e., when the second observation becomes available. The top row represents the history $\mathcal{H}_t$, the second row represents the hypernetwork recurrence, the third row represents the latent Hawkes process, and the bottom row are the intensities. Note we suppress the arrow from $t_1$ into $\mathbf{x}_{2-}$ for visual clarity.

To predict $\mathbf{D}_i$, we pass $\mathbf{z}_i$ through a learned linear transforms to produce $\mathbf{d}_i \in \mathbb{R}^d$. From there, we compute $\mathbf{D}_i = -\text{diag}(\text{softplus}(\mathbf{d}_i) \odot \mathbf{u})$ where $\odot$ is an element-wise product and $\mathbf{u} \in \mathbb{C}^d$ with $\log \Re(\mathbf{u}) \in \mathbb{R}^d$. Similarly, to predict $\mathbf{V}_i$, we pass $\mathbf{z}_i$ through a separate linear transform to produce $\mathbf{v}_i \in \mathbb{R}^{2dr}$ where $r$ is a hyperparameter. These values become angles through which to parameterize a unitary matrix as described by Jing et al. (2017). Here, $r$ determines the number of $\mathbf{F}$-component matrices as denoted in their work. Following their parameterization produces $\mathbf{V}_i$, where matrix-vector products reduce to a sequence of component-wise vector multiplications and vector permutations.

This completes the iteration, with a new hypernetwork state and latent Hawkes state ready for the next iteration.

**Computational Complexity.** Both the GRU-based hypernetwork update and the Hawkes recurrence have constant time and memory complexity, $\mathcal{O}(1)$, making inference highly scalable with respect to sequence length.

## A.2 ARCHITECTURE HYPERPARAMETERS AND LEARNABLE PARAMETERS

The hypernetwork depends solely on the event history and emits the parameters $\mathbf{V}_i$ and $\mathbf{D}_i$ for the Hawkes recurrence. We use a GRU combined with deterministic orthonormal matrix construction (Jing et al., 2017). The GRU state $\mathbf{z}_i$ takes as input the logarithm of the time since the last event and a mark embedding. This embedding dimension is a hyperparameter, set to one less than the GRU state dimension to maintain consistent input size. The GRU parameters $\phi \in \Phi$ include its initial state.

The learnable parameters of the HHP are therefore:

- GRU hypernetwork parameters $\phi \in \Phi$,
- Mark-specific impulses $\boldsymbol{\alpha} \in \mathbb{R}^{d \times K}$,
- Emission layer parameters: projection matrix $\mathbf{W} \in \mathbb{R}^{K \times d}$ and background intensity $\boldsymbol{\mu} \in \mathbb{R}^K$.

The full parameter set is:
$$\theta = \{\phi, \boldsymbol{\alpha}, \mathbf{W}, \boldsymbol{\mu}\} \in \Theta.$$

Key architectural hyperparameters include:

- Latent dimension of the Hawkes process ($d$),
- Dimension of the hypernetwork recurrence (we use a GRU and do not explore alternatives here).

## A.3 COMPUTING THE LOG-LIKELIHOOD

The log-likelihood for intensity-based MTPPs is defined in Equation (2), with further background in Daley and Vere-Jones (2003). At a high level:

1. For a given event sequence, we compute the left-limit intensities for each observed mark type $k_i$, denoted $\left\{\lambda_{t_i}^{k_i}\right\}_{i=1}^{L}$, using the procedure in Section A.1.

2. These intensities form the first term of the log-likelihood.

3. To approximate the normalizing integral, we sample a fixed number of points $t' \in (t_i, t_{i+1})$ uniformly and compute the total intensity $\lambda_{t'} = \sum_{k=1}^{K} \lambda_{t'}^{k}$ at each sampled time.

Importantly, the GRU recurrence is computed only once per event, not per sample point, since it is conditioned solely on events. This allows us to amortize its cost across all sampled points.

**Computational Complexity.** Due to the conditional linearity of the Hawkes recurrence, it can be computed in logarithmic time $\mathcal{O}(\log L)$ using parallel scans (Chang et al., 2025), assuming sufficient computational resources. The evaluation of all sampled points for the normalizing constant can be done in constant time $\mathcal{O}(1)$, as they are conditionally independent given the recurrence right limits.

The main computational bottleneck is the sequential nature of the GRU hypernetwork. If training throughput is critical, this could be mitigated by adopting parallelizable sequence models such as self-attention (Vaswani et al., 2017), deep state space models (Gu and Dao, 2023), or parallelization techniques for non-linear recurrent sequence models (Lim et al., 2023; Gonzalez et al., 2024).

# B    ADDITIONAL RESULTS & EXPERIMENT DETAILS

This appendix provides additional experimental details and results for all models evaluated in this work, including our proposed Hyper Hawkes Process (HHP). All experiments were conducted in a unified environment, using identical data splits, pre-processing, and evaluation protocols for both HHP and baseline models. No additional pre-processing or special training procedures were required for HHP beyond what was used for prior models.

## B.1    TRAINING DETAILS & HYPERPARAMETER CONFIGURATIONS

For all baseline models, we use the hyperparameters and architectures as reported in Chang et al. (2025). For HHP, we performed a grid search over latent dimension $d$ (values: $\{8, 16, 32, 64, 128, 256\}$), GRU hidden size $h$ ($\{16, 32, 64, 128, 256\}$), GRU layers $l$ ($\{1, 2\}$), and number of rotation matrices used in $\mathbf{V}_i$ parameterization $r$ ($\{2, 4, 8\}$). The chosen values for each dataset are reported in Table 3.

Table 3: Chosen hyperparameters for HHP across all seven benchmark datasets.

| Dataset | $d$ | $h$ | $l$ | $r$ | # Parameters |
|---|---|---|---|---|---|
| Amazon | 64 | 8 | 2 | 8 | 11240 |
| Retweet | 64 | 32 | 2 | 8 | 23940 |
| Taxi | 128 | 8 | 2 | 8 | 9656 |
| Taobao | 64 | 8 | 2 | 4 | 5104 |
| StackOverflow | 64 | 8 | 2 | 8 | 6936 |
| Last.fm | 64 | 32 | 2 | 8 | 42777 |
| MIMIC-II | 256 | 16 | 2 | 8 | 126336 |

## B.2    DATASET STATISTICS

We report the statistics of all seven datasets used in this work in Table 4. We used the `HuggingFace` version of the five `EasyTPP` datasets. For all datasets, we ensured that no more than two events occur at the same time (i.e., inter-arrival time is strictly positive), and event times do not lie on grid points that are effectively discrete-time events. Dataset descriptions and pre-processing details are provided in Appendix B.3.

Table 4: Statistics of the seven datasets we experiment with.

| Dataset | $K$ | Number of Events | | | Sequence Length | | | Number of Sequences | | |
|---|---|---|---|---|---|---|---|---|---|---|
| | | Train | Valid | Test | Min | Max | Mean | Train | Valid | Test |
| Amazon | 16 | 288,377 | 40,995 | 84,048 | 14 | 94 | 44.8 | 6,454 | 922 | 1,851 |
| Retweet | 3 | 2,176,116 | 215,521 | 218,465 | 50 | 264 | 108.8 | 20,000 | 2,000 | 2,000 |
| Taxi | 10 | 51,584 | 7,404 | 14,820 | 36 | 38 | 37.0 | 1,400 | 200 | 400 |
| Taobao | 17 | 73,483 | 11,472 | 28,455 | 28 | 64 | 56.7 | 1,300 | 200 | 500 |
| StackOverflow | 22 | 90,497 | 25,762 | 26,518 | 41 | 101 | 64.8 | 1,401 | 401 | 401 |
| Last.fm | 120 | 1,534,738 | 344,542 | 336,676 | 6 | 501 | 207.2 | 7,488 | 1,604 | 1,604 |
| MIMIC-II | 75 | 9,619 | 1,253 | 1,223 | 2 | 33 | 3.7 | 2600 | 325 | 325 |

## B.3    DATASET PRE-PROCESSING

We used the default train/validation/test splits for the `EasyTPP` benchmark datasets. For MIMIC-II, we followed Du et al. (2016) and kept the 325 test sequences in the test split, further splitting the 2,935 training sequences into 2,600 for training and 325 for validation. For Last.fm, we randomly partitioned the data into 70%, 15%, and 15% splits for training, validation, and test, respectively. For all datasets, a small amount of jitter was added to event times if necessary to ensure no two events occurred at the same time and to avoid discrete-time artifacts.

**Amazon** (Ni et al., 2019) contains user product reviews, with product categories as marks. **Retweet** (Zhao et al., 2015) models retweet cascades, with event types based on user influence.

Table 5: Full breakdown of log-likelihood metrics.

| Model | Per Event Log-Likelihood, $\mathcal{L}_{\text{Total}}$ (nats) ($\uparrow$) | | | | | | | Avg. Ranking ($\downarrow$) |
|---|---|---|---|---|---|---|---|---|
| | Amazon | Retweet | Taxi | Taobao | StackOverflow | Last.fm | MIMIC-II | |
| RMTPP | -2.136 (0.003) | -7.098 (0.217) | 0.346 (0.002) | 1.003 (0.004) | -2.480 (0.019) | -1.780 (0.005) | -0.472 (0.026) | 7.3 |
| NHP | 0.129 (0.012) | **-6.348** (0.000) | 0.514 (0.004) | 1.157 (0.004) | -2.241 (0.002) | -0.574 (0.011) | 0.060 (0.017) | 4.0 |
| SAHP | -2.074 (0.029) | -6.708 (0.029) | 0.298 (0.057) | 1.168 (0.029) | -2.341 (0.058) | -1.646 (0.083) | -0.677 (0.072) | 6.6 |
| THP | -2.096 (0.002) | -6.659 (0.007) | 0.372 (0.002) | 0.790 (0.002) | -2.338 (0.014) | -1.712 (0.011) | -0.577 (0.011) | 6.6 |
| IFTPP | 0.496 (0.002) | -10.344 (0.016) | 0.453 (0.002) | **1.318** (0.017) | -2.233 (0.009) | **-0.492** (0.017) | 0.317 (0.052) | 3.6 |
| AttNHP | 0.484 (0.077) | -6.499 (0.028) | 0.493 (0.009) | 1.259 (0.022) | -2.194 (0.016) | -0.592 (0.051) | -0.170 (0.077) | 3.9 |
| S2P2 | **0.781** (0.011) | -6.365 (0.003) | **0.522** (0.004) | 1.304 (0.039) | **-2.163** (0.009) | -0.557 (0.046) | 0.919 (0.069) | **1.7** |
| HHP (ours) | 0.616 (0.016) | -6.366 (0.007) | 0.520 (0.003) | 1.232 (0.014) | -2.209 (0.006) | -0.515 (0.006) | **1.314** (0.048) | 2.4 |

| Model | Per Event Next Event Time Log-Likelihood, $\mathcal{L}_{\text{Time}}$ (nats) ($\uparrow$) | | | | | | | Avg. Ranking ($\downarrow$) |
|---|---|---|---|---|---|---|---|---|
| | Amazon | Retweet | Taxi | Taobao | StackOverflow | Last.fm | MIMIC-II | |
| RMTPP | 0.011 (0.001) | -6.191 (0.083) | 0.622 (0.002) | 2.428 (0.004) | -0.797 (0.005) | 0.256 (0.007) | -0.188 (0.016) | 7.0 |
| NHP | 2.116 (0.009) | **-5.584** (0.001) | 0.727 (0.003) | 2.578 (0.006) | -0.699 (0.002) | 1.198 (0.006) | 0.225 (0.016) | 4.3 |
| SAHP | 0.115 (0.049) | -5.872 (0.062) | 0.645 (0.044) | 2.604 (0.008) | -0.703 (0.031) | 0.489 (0.078) | -0.244 (0.040) | 5.9 |
| THP | -0.068 (0.002) | -5.874 (0.007) | 0.621 (0.002) | 2.242 (0.002) | -0.772 (0.006) | 0.220 (0.010) | -0.271 (0.004) | 7.6 |
| IFTPP | 2.483 (0.001) | -9.500 (0.011) | **0.735** (0.002) | 2.708 (0.018) | -0.662 (0.007) | **1.277** (0.016) | 0.555 (0.050) | 2.9 |
| AttNHP | 2.416 (0.092) | -5.726 (0.027) | 0.714 (0.010) | 2.654 (0.007) | -0.684 (0.005) | 1.203 (0.015) | 0.031 (0.055) | 4.1 |
| S2P2 | **2.652** (0.009) | -5.598 (0.002) | 0.733 (0.003) | **2.719** (0.038) | **-0.641** (0.003) | 1.257 (0.022) | 1.050 (0.065) | **1.7** |
| HHP (ours) | 2.492 (0.015) | -5.597 (0.005) | 0.732 (0.002) | 2.620 (0.013) | -0.670 (0.002) | 1.252 (0.004) | **1.394** (0.043) | 2.6 |

| Model | Per Event Next Mark Log-Likelihood, $\mathcal{L}_{\text{Mark}}$ (nats) ($\uparrow$) | | | | | | | Avg. Ranking ($\downarrow$) |
|---|---|---|---|---|---|---|---|---|
| | Amazon | Retweet | Taxi | Taobao | StackOverflow | Last.fm | MIMIC-II | |
| RMTPP | -2.147 (0.003) | -0.908 (0.141) | -0.276 (0.000) | -1.425 (0.002) | -1.683 (0.015) | -2.035 (0.004) | -0.284 (0.014) | 6.9 |
| NHP | -1.987 (0.003) | **-0.764** (0.000) | -0.213 (0.002) | -1.421 (0.004) | -1.542 (0.001) | -1.772 (0.006) | -0.165 (0.002) | 3.3 |
| SAHP | -2.189 (0.030) | -0.836 (0.036) | -0.346 (0.024) | -1.436 (0.027) | -1.638 (0.032) | -2.136 (0.070) | -0.433 (0.031) | 7.4 |
| THP | -2.028 (0.002) | -0.785 (0.001) | -0.249 (0.001) | -1.451 (0.000) | -1.566 (0.008) | -1.932 (0.006) | -0.306 (0.009) | 6.0 |
| IFTPP | -1.988 (0.001) | -0.844 (0.007) | -0.282 (0.001) | -1.391 (0.005) | -1.571 (0.003) | -1.769 (0.004) | -0.239 (0.002) | 4.9 |
| AttNHP | -1.933 (0.024) | -0.773 (0.003) | -0.221 (0.002) | -1.395 (0.016) | **-1.510** (0.013) | -1.795 (0.037) | -0.201 (0.025) | 3.3 |
| S2P2 | **-1.871** (0.002) | -0.767 (0.000) | **-0.211** (0.002) | -1.415 (0.005) | -1.521 (0.008) | -1.814 (0.025) | -0.131 (0.014) | 2.4 |
| HHP (ours) | -1.877 (0.002) | -0.769 (0.002) | -0.212 (0.001) | **-1.388** (0.003) | -1.539 (0.004) | **-1.767** (0.003) | **-0.079** (0.009) | **1.9** |

**Taxi** (Whong, 2014; Mei et al., 2019) uses New York taxi pickup/dropoff data, with marks defined by location-action pairs. **Taobao** (Xue et al., 2022) consists of e-commerce viewing patterns, with item categories as marks. **StackOverflow** contains badges awarded to users on a Q&A website, with badge type as the mark. **MIMIC-II** (Saeed et al., 2002) records disease events during hospital visits, with disease type as the mark. For MIMIC-II and StackOverflow, we used the pre-processing from Du et al. (2016). **Last.fm** (Celma Herrada et al., 2009; McFee et al., 2012) records music listening habits, with genres as marks. Each event is a play of a particular genre, and if a song had multiple genres, one was selected at random.

## B.4 FULL RESULTS ON BENCHMARK DATASETS

We provide the full log-likelihood results in Table 5, decomposing likelihood into time and mark components. Our HHP model achieves strong performance across all metrics, with improvements primarily driven by better modeling of event times. HHP also achieves best- or second-best accuracy for next mark prediction on most datasets. Likewise, time and mark calibration results, as measured by PCE and ECE, respectively, can be found in Table 6. We implement these metrics as defined by Bosser and Taieb (2023). In this aspect, we see that our model performs similarly to the baseline methods, with most being reasonably well-calibrated.

Table 6: Calibration results for the models and datasets tests.

(a) Probabilistic calibration error (PCE) for time calibration in percentage.

| Model | Probabilistic Calibration Error (PCE) (↓) | | | | | | | Average Ranking (↓) | [Added] Average PCE (↓) |
|---|---|---|---|---|---|---|---|---|---|
| | Amazon | Retweet | Taxi | Taobao | StackOverflow | Last.fm | MIMIC-II | | |
| RMTPP | 13.67 (0.03) | 7.93 (0.62) | 3.50 (0.03) | **0.22** (0.16) | 1.94 (0.10) | 1.56 (0.01) | 3.63 (0.37) | 5.6 | 4.64 |
| NHP | 8.45 (0.28) | **0.20** (0.19) | 0.87 (0.50) | 7.40 (0.68) | 1.51 (0.11) | 4.70 (0.13) | 5.92 (0.14) | 5.6 | 4.15 |
| SAHP | 12.04 (1.02) | 8.51 (1.86) | 2.52 (0.99) | 3.18 (0.21) | 1.50 (0.57) | 2.53 (1.86) | 2.28 (0.44) | 5.4 | 4.65 |
| THP | 12.38 (0.05) | 5.68 (0.08) | 3.34 (0.02) | 6.36 (0.04) | 2.06 (0.11) | 1.02 (0.08) | **1.10** (0.06) | 5.3 | 4.56 |
| IFTPP | **1.59** (0.09) | 23.85 (0.26) | **0.40** (0.10) | 1.61 (0.74) | **0.84** (0.34) | **0.46** (0.44) | 1.75 (0.33) | **2.3** | 4.36 |
| AttNHP | 6.36 (0.63) | 2.09 (0.85) | 0.84 (0.27) | 3.08 (0.16) | 1.65 (0.24) | 1.43 (0.14) | 4.70 (0.33) | 4.6 | 2.88 |
| S2P2 | 5.88 (0.17) | 0.44 (0.27) | 0.55 (0.33) | 2.07 (0.32) | 1.03 (0.15) | 1.38 (0.52) | 11.70 (0.68) | 3.4 | 3.29 |
| HHP (ours) | 6.74 (0.54) | 0.59 (0.47) | 0.43 (0.18) | 2.97 (0.62) | 1.01 (0.24) | 2.91 (0.38) | 4.25 (1.45) | 3.9 | **2.69** |

(b) Expected calibration error (ECE) for mark calibration in percentage.

| Model | Expected Calibration Error (ECE) (↓) | | | | | | | Average Ranking (↓) | [Added] Average ECE (↓) |
|---|---|---|---|---|---|---|---|---|---|
| | Amazon | Retweet | Taxi | Taobao | StackOverflow | Last.fm | MIMIC-II | | |
| RMTPP | 6.58 (0.15) | 3.99 (4.28) | 2.42 (0.16) | 1.89 (0.24) | 2.10 (0.27) | 2.47 (0.45) | 2.79 (0.43) | 6.0 | 3.18 |
| NHP | 8.30 (0.21) | **0.35** (0.06) | 0.79 (0.10) | 5.59 (0.69) | 1.31 (0.16) | 3.41 (0.41) | 2.24 (0.32) | 5.0 | 3.14 |
| SAHP | 8.17 (2.00) | 6.27 (2.23) | 6.77 (0.21) | 2.68 (0.35) | 1.71 (0.77) | 6.26 (4.30) | 5.41 (0.26) | 7.1 | 5.32 |
| THP | 2.06 (0.17) | 1.26 (0.11) | 1.76 (0.07) | 6.51 (0.03) | **0.81** (0.14) | 3.42 (0.70) | 2.16 (0.39) | 5.1 | 2.57 |
| IFTPP | **0.46** (0.10) | 0.95 (1.12) | **0.55** (0.19) | **1.20** (0.20) | 1.28 (0.54) | 0.66 (0.05) | **1.39** (0.23) | **1.9** | **0.93** |
| AttNHP | 3.13 (0.61) | 0.52 (0.16) | 0.56 (0.10) | 2.47 (0.12) | 1.37 (0.42) | **0.61** (0.16) | 2.23 (0.50) | 3.6 | 1.56 |
| S2P2 | 0.88 (0.34) | 0.52 (0.13) | 0.58 (0.12) | 1.96 (0.67) | 1.98 (0.19) | 1.01 (0.63) | 1.62 (0.24) | 3.4 | 1.22 |
| HHP (ours) | 1.53 (0.22) | 0.38 (0.38) | 0.83 (0.09) | 1.91 (0.29) | 2.00 (0.64) | 1.44 (0.53) | 1.54 (0.33) | 3.7 | 1.38 |

## C INTERPRETABILITY EXPLORATION

In this appendix, we provide more concrete details on the interpretability scenario explored in the main paper, as well as introduce another scenario—along with an accompanying analysis using the proposed interpretability tools.

### C.1 SCENARIO FROM SECTION 6.1

**Data Generating Process.** Sequences were sampled one event at a time, being drawn from a Poisson process with rate $\lambda = 1/3$. Marks are then randomly assigned to these events with probability $40\%$ for blue, $40\%$ for orange, and $20\%$ for green. Should a green event be drawn at time $t$, we denote that as a "trigger" event. The immediate next event that is drawn will have the exact same mark as the event that came before the trigger, and the time of the event will be drawn from $t + \mathcal{N}(10, 0.01)$. After this follow-up event is drawn, we return to drawing from the Poisson process as before. A sequence is done sampling once we reach $T = 100$. See Appendix C.1 for example sequences generated under this process.

An HHP model was trained on 2,000 generated sequences, with a latent dimension of $d = 32$, a hidden dimension of $h = 8$ for a single-layered GRU, and only $r = 2$ predicted component blocks for the eigenvectors. The resulting model possesses 1328 parameters. The rest of the training details, e.g., epochs, batch-size, etc., are identical to the main set of experiments.

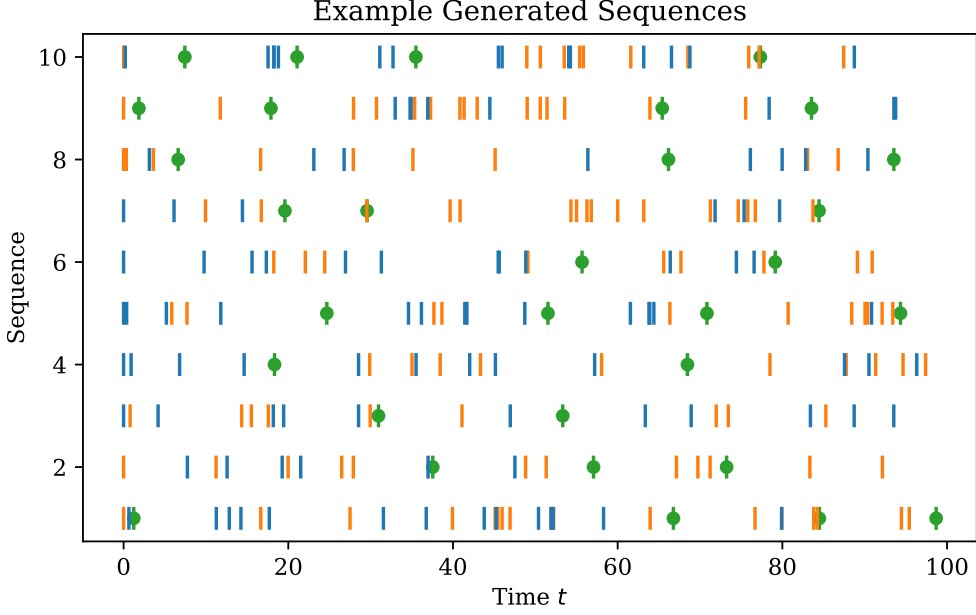

Figure 4: Visualization of ten example sequences drawn from the data generating process that was analyzed in Section 6.1. Trigger events are overlaid with dots for better readability.

## C.2 SECOND SCENARIO

**Data Generating Process.** For this new scenario, we will simulate two processes separately and then treat the superposition of them as a single sequence to model. The first process is simple, green events are drawn from a Poisson process at rate $\lambda = 1/2$. For the second process, we will simulate a sequence of pairs of call and response events. We will label a "call" event as blue and draw it from an exponential distribution with rate $\lambda = 1/15$. After drawn, the "response" event, which we will denote as orange, has its time equal to the call event plus a random offset drawn from $\mathcal{N}(10, 0.01)$. Afterwards, another call event is drawn offset from the previous response with the same exponential distribution as before, and so on. The superposition of the two produces a sequence with three possible marks, spanning $t \in [0, 100]$. See Appendix C.2 for example sequences generated under this process.

We trained an HHP model on 2,000 generated sequences from this process. The rest of the training setup is identical to the previous synthetic scenario.

**Aggregate Statistics.**

While the interpretability of HHP is uniquely suited towards event-level attribution, marginal effects are still possible. These can be achieved by aggregating the leave-one-out estimators across multiple events and sequences. For instance, we can get a broad sense of how the model chooses to leverage particles of various types by understanding the general distribution of the total influence these particles have on the output. This can be measured on a per-event basis via $\sum_{m=1}^{K} \mathrm{DF}|\Lambda_T^m|$ where $T$ is the length of the time window. This value describes the total influence that a given particle has had over its entire lifetime and is measured on the scale of number of events.

Appendix C.2 shows the distribution of these lifetime influences grouped by the events' marks. We can see that in general, green events are quickly discarded by the model as they do not have much lasting influence over future events. This makes sense given that these were generated by a background process and have no influence over other events. Conversely, the call (blue) events are shown to have a stronger influence over their lifetime, averaging a total influence of roughly 2 events. Since we know that the true data generating process will alternate call and response events in

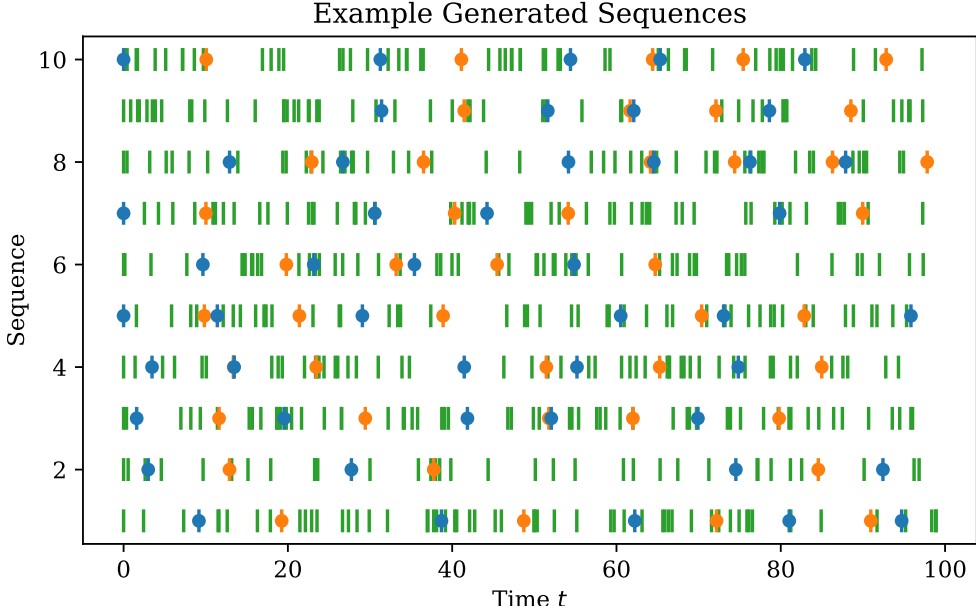

Figure 5: Visualization of ten example sequences drawn from the data generating process in the second synthetic scenario. Call (blue) and response (orange) events are overlaid with dots for better readability. Note that a response event can only occur after a call event has happened, and vice versa, regardless of how many or few green events occur in the interim.

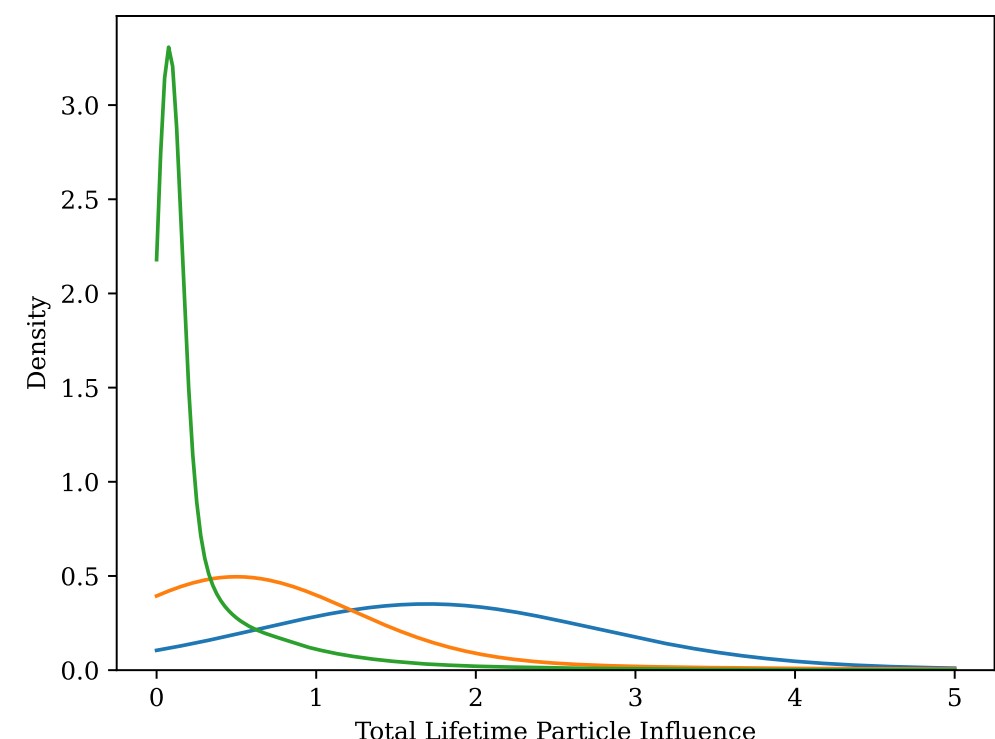

Figure 6: Displayed is the density of lifetime total influence $\sum_{m=1}^{K} \mathrm{DF}|\Lambda_T^m|$ that individual particles have on the model's predictions, aggregated over every event from all sequences in the generated data. Each density plotted corresponds to the particle's mark.

a ratio of 1:1, it appears that the model leverages these particles past the subsequent response event prediction. This is mirrored by the response events having more influence than the green events but still less than 1 event on average. To better understand this, we will dive deeper and analyze how the model responds to events from a single sequence.

**Output Intensities.**

We have chosen a held-out sequence chosen at random from the data generating process. Appendix C.2 shows the resulting predicted intensities that the HHP model produces when conditioned on the sequence. We can see that when a call (blue) event occurs, the intensity for both call and response (orange) events drop to near zero (but never exactly zero due to $\sigma(z) \in (0, \infty)$). These values remain there until about 10 units of time later when the response intensity spikes. Then after a response event occurs, the intensities reset back to normal. All the while the green intensity is roughly stable and unresponsive to any ongoing events, which mirrors the true generation process. For the remainder of this section, this sequence will be used for subsequent analysis.

**Individual Effects.**

Now that we have selected a sequence and observed the overall output intensity from the model, we can dive deeper and understand how each event's particle is being used by the model to influence the output. In Appendix C.2, we plot for each particle in the sequence the entire trace of $\mathrm{DF}\lambda_t^{1:K}$ for $t \in [0, T]$. This showcases the first-order effects that the particle has over time on each output marked intensity for the model.

There are a number of interesting effects and patterns that give us a glimpse into how the model is choosing to arrive at its predictions. First, we can see that most of the events leading up to the first response (orange) event all appear to be leveraged by the model to spike and excite a response event to occur. Strangely, there are a couple of events that are also used to inhibit the response event just before it occurs as well. This indicates that in latent space there is likely a complex push-pull

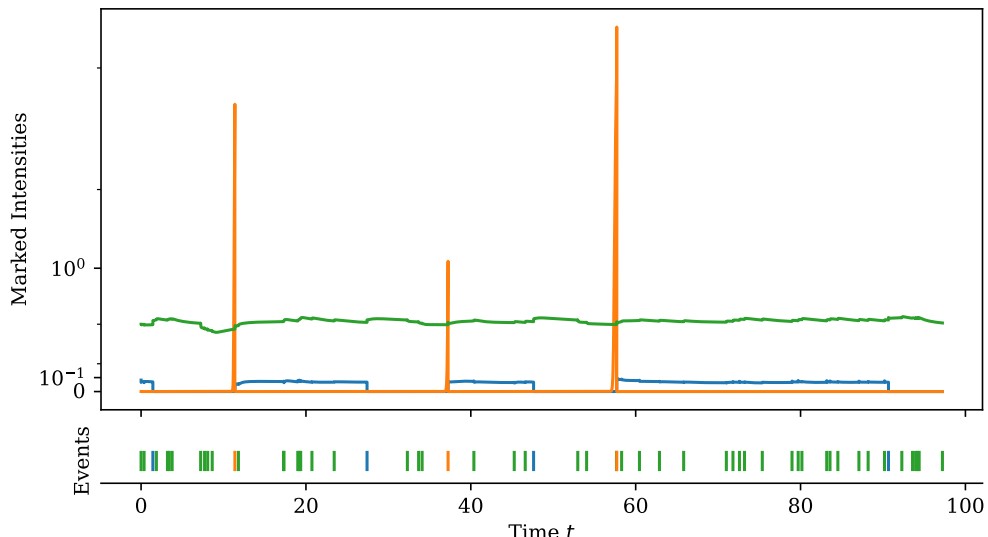

Figure 7: On top are the model's predicted intensities, each line corresponding to a mark matching in color. On bottom is the corresponding sequence the model is conditioning on to produce the above intensities. Note that when a blue event occurs, the orange and blue intensities drop to near zero, only to have an orange intensity spike around 10 units of time later.

between particles in an attempt to arrive a what we know to be a correctly timed, large excitation for the response event, as indicated in Appendix C.2.

After this first response event occurs, we can see the particles are effectively killed thereafter as they have little to no influence moving forward. In a way, it is as if the model has reset at this point. Resetting after a response event does not always seem to be the case though for the rest of the sequence. We can see some particles contribute to spiking for both of the last two response events in the sequence. When debugging a model, insights like this can help give inspiration for attempting new mechanisms to help guide these various behaviors, like resetting.

**Joint Effects.**

While this is a large amount of information available to mine, it is important to note that these signals pertain to *per-event* effects. They are just analyzing how the model outputs would differ if that single particle were not present; however, we know that many particles can interact in latent space and produce greater effects than just the sum of their individual effects. To this end, we can also measure higher-order effects, as mentioned in the main paper.

In Appendix C.2, we showcase a heatmap of the interaction effects of pairs of particles in an attempt to visualize how "coupled" a pair is. This is measured as the absolute difference in total joint lifetime influence of a pair of events, $\mathrm{DF}|\Lambda_T^{(i,j)}|$, and the naive first-order estimation of this effect, $\mathrm{DF}|\Lambda_T^{(i)}|+\mathrm{DF}|\Lambda_T^{(j)}|$. This can be thought of as the DFBETA for a linear regression model's interaction term, i.e., measuring effect of $\beta_{12}$ by comparing $\hat{y} = \beta_1 x_1 + \beta_2 x_2$ to $\hat{y} = \beta_1 x_1 + \beta_2 x_2 + \beta_{12} x_1 x_2$. While the resulting scale is on the order of number of events, it should be noted that this does not measure how strong the influence a pair of events is, but rather just how much do the two particles interact with one another.

In the figure we can see interesting patterns emerge. Namely, we see strong interaction effects between the call (B) / response (O) events and all other events, as indicated by the highlighted columns and rows. From this, we know that the model is not choosing dynamics that move these individual particles in isolation, but rather are positioning them contextually amongst all other particles and relying on them to constructively or destructively interfere with one another w.r.t. the output intensities. Additionally, we can see a large bright spot in the middle of the heatmap near where two pairs of call and response events occurred right after one another. This correlates with the

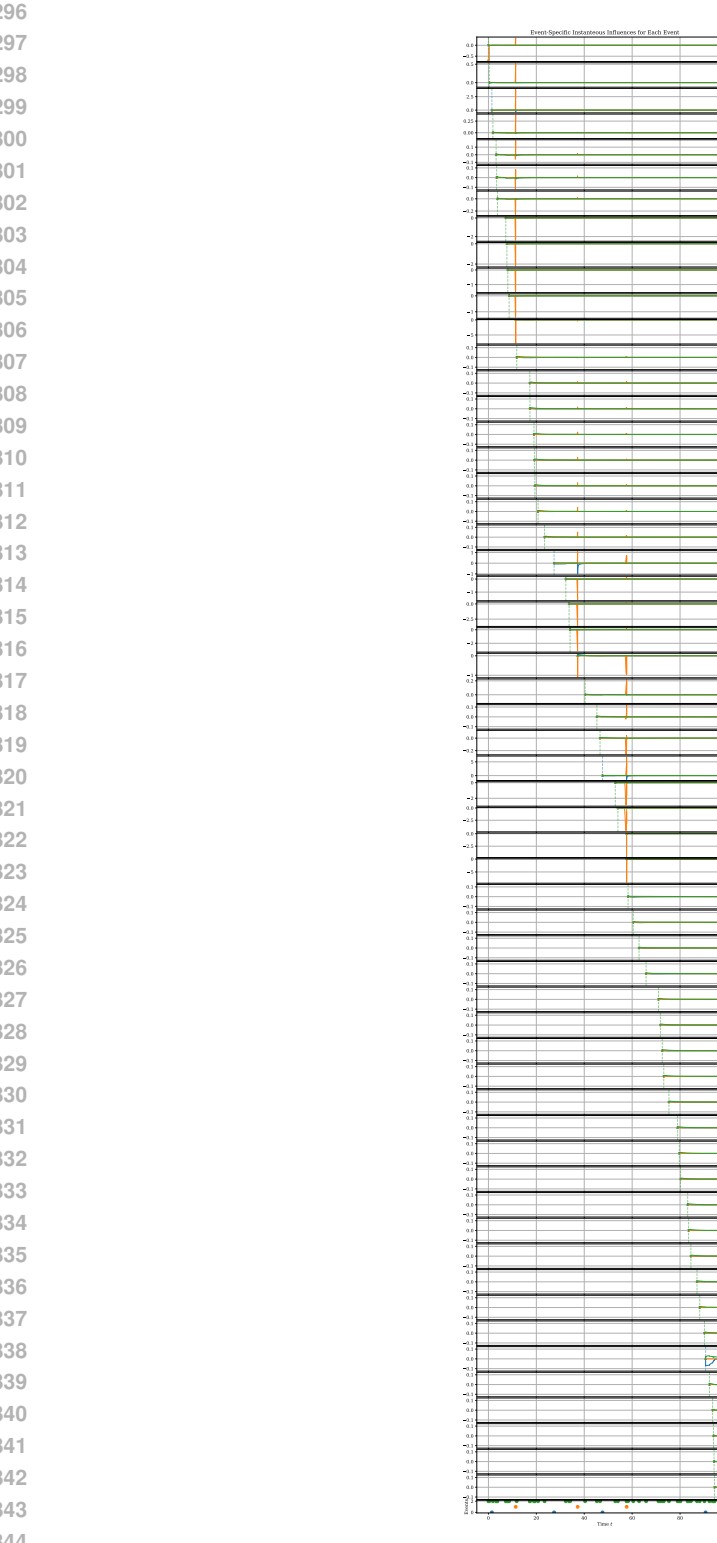

Figure 8: **Please view as a PDF to zoom in to details.** For the same sequence showcased in Appendix C.2, the individual $\mathrm{DF}\lambda_t^{(i)}$ values over time for a particle are displayed with the top-most plot showing the first event, $i = 1$, and second-to-last showing the last event, $i = N_T$. The color of the dashed line in each subplot indicates the mark of the particle being displayed. The colors of the solid lines indicate the instantaneous influence that particle has over future events of that color.

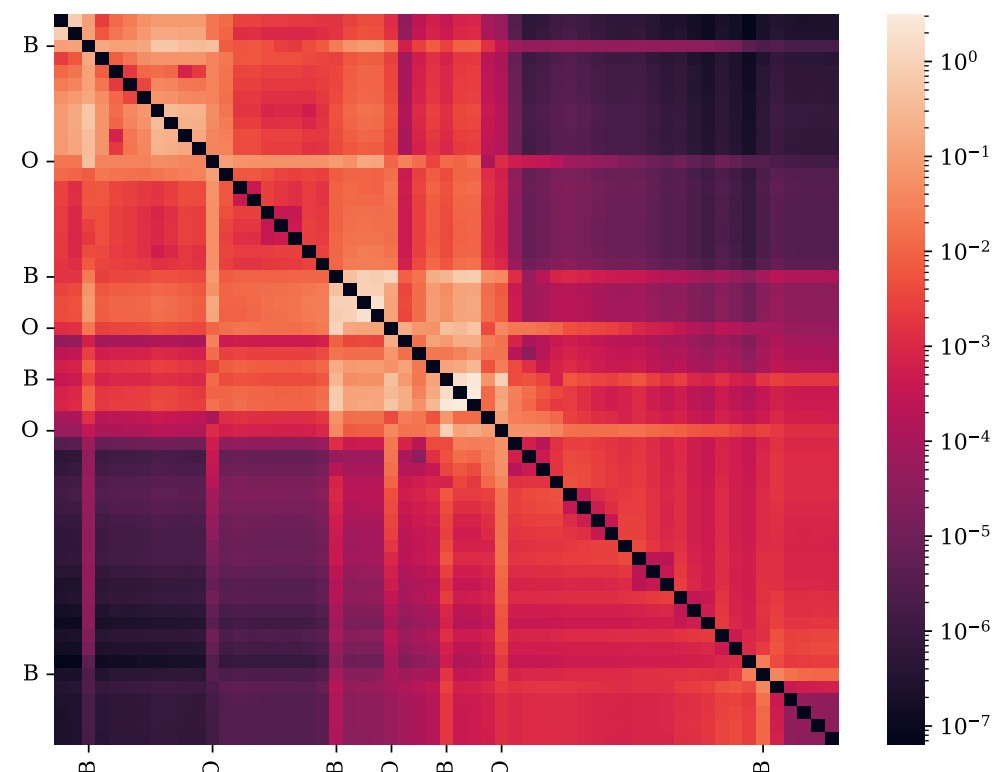

Figure 9: A heatmap measuring how "coupled" pairs of events are for the sequence showcased in Appendix C.2. Events are ordered as first to last from top to bottom and from left to right. Call (blue) and response (orange) events are labeled with 'B' and 'O', respectively, with green events having no label. Higher values indicate tighter coupling and values close to zero indicate no coupling.

individual effects we saw earlier in that there were particles that were leveraged for multiple response excitations.

**Retrospective Attribution.**

Lastly, it is worth demonstrating that the proposed tools can also be used to pinpoint specific information. To that end, we will showcase one view into what events contribute towards the occurrence of a specific event. Put differently, given that an event occurred, how much did each prior event either excite or inhibit that occurrence?

To measure this, say that the specific event in question is the $i^{\text{th}}$ event that occurred at time $t_i$ with mark $k_i$. The influence, positive or negative, that the $j^{\text{th}}$ event for $j < i$ has is measured by $\text{DF}\lambda_{t_i}^{k_i(j)}$. We have shown this breakdown for the three response (orange) events and the influence that all events prior to them had. From this view, it becomes apparent the strong influence that the call (blue) events have on the response, and specifically the most recent call events. These values in this perspective can be roughly treated as attention scores; however, the scale of them is on the same order of intensities so the magnitude is meaningful. Additionally, unlike attention in multi-layer transformers, these statistics were derived from the linear recurrence bottleneck for HHP, which makes these values clearly tied to the events they represent.

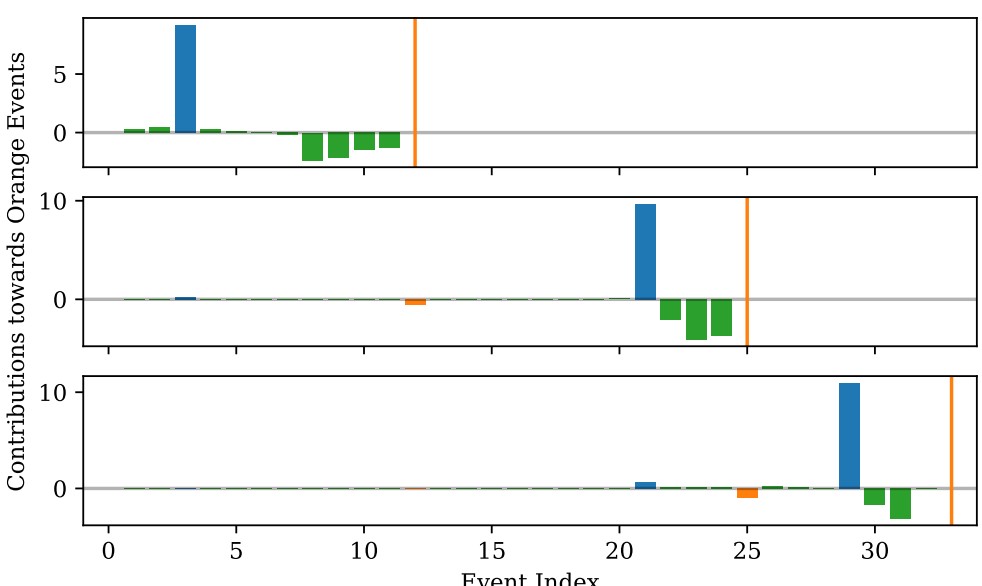

Figure 10: Retrospective attributions for the three response (orange) events from the sequence showcased in Appendix C.2, with the top corresponding to the first response event and the bottom to the last. Bars indicate the instantaneous contributions that a prior event had towards the response event. More exactly, the $i^{\text{th}}$ bar displays $\mathrm{DF}\lambda_{t_j}^{k_j(i)}$ where $j$ is the index of the response event. Bars are colored by the corresponding event's mark $k_i$.

