# OpenReview forum: "Hyper Hawkes Processes: Interpretable Models of Marked Temporal Point Processes"
_ICLR.cc/2026/Conference — Submitted to ICLR 2026_

### Official Review · Reviewer_gUR2 · 2025-10-30

**Soundness:** 2
**Presentation:** 2
**Contribution:** 2
**Rating:** 4
**Confidence:** 4

**Summary:**

This paper introduces the Hyper Hawkes Process (HHP), a novel marked temporal point process model that combines the interpretability of classical Hawkes processes with the expressivity of neural MTPPs. The key innovations include: (1) lifting Hawkes dynamics into a latent space to decouple from mark dimensionality, (2) employing a hypernetwork to generate time-varying dynamics while maintaining conditionally linear recurrence, and (3) developing event-level attribution methods for interpretability. The paper also demonstrates competitive performance across seven real-world benchmarks and provide interpretability analysis on synthetic tasks.

**Strengths:**

(1). The idea of combining a hypernetwork with a linear Hawkes recurrence is well-motivated.

(2). The model in the paper is also rigorously developed, with clear motivations for each extension (latent space, time-varying dynamics, etc.).

(3). The work contributes to both the performance and interpretability of MTPPs, which is important for real-world applications (e.g., healthcare, finance)

**Weaknesses:**

(1). The paper should better highlight how HHP differs from and improves upon these models.

(2). The interpretability tools are only demonstrated on synthetic data. It's better to also show their utility on real-world datasets.

(3). The choice of a GRU-based hypernetwork is not justified against alternatives such as transformers, SSMs and so on.

**Questions:**

(1). Why was a GRU chosen over other sequence models? Was attention or Mamba considered?

(2). Can the authors discuss how HHP could be used in a real-world setting (e.g., healthcare) to provide insights beyond prediction?

**Details Of Ethics Concerns:**

It would be better to use the medical (MIMIC-II)  datasets accompanied by a brief discussion of potential biases and the responsible use of interpretable models in sensitive domains.

---

> ### Author Response · Authors · 2025-11-22
>
> Firstly, thank you to the reviewer for taking the time to review and comment on our paper.  We were very pleased with the reviewers comments highlighting the clear motivation and rigor in the development and presentation our model!  In regard to your weaknesses and questions:
>
> - `(1). The paper...`:  This is a very fair point, on reflection we could have done a better job comparing and contrasting our architecture with existing architectures.  For any camera ready version, we will use the extra page to increase the discussion and trade-offs with existing architectures.
>
> - `(3). The choice...` & `(1). Why was...`:  This is a great point for clarification.  We initially selected the GRU because of its widely available implementation and robust performance. Interestingly, we do not actually view the GRU specifically as a fixed part of the HHP architecture, which is maybe why this was under-discussed in the submission. Any non-linear causal model could be dropped-in in place of the GRU.  We have added clarification of this.
>
> - `(2). Can the...`:  This is fascinating as a topic!  We are very happy that the HHP opens these up as avenues for discussion.  The event-level attribution is particularly relevant for healthcare data, where _comorbitities_ are so important for predicting the onset of future conditions.  For instance, attribution could identify which combination of prior diagnoses and when most influence risk across time. This sort of analysis is facilitated through the ``Group Influence'' discussed in Section 6.1 (near line 393) and in (e.g.) Figure 9.  This simply cannot be answered with existing black-box neural methods (without incurring combinatorial computational cost).  We will highlight this in the discussion section.
>
> - In regard to the highlighting of an Ethical Concern:  We commit to adding a discussion of the risks and harms of interpretable methods on healthcare data.  We do highlight our method is no more or less susceptible to this than any other interpretable method, and, the MIMIC-II data used in particular has had all PHI stripped.  However, we agree it does warrant explicit discussion, and will add a dedicated subsection to the Discussion.  As a side-note, the lead authors have completed the "CITI Data or Specimens" handling training, concerning the use and protections for healthcare data required for the original MIMIC dataset.
>
>
>
> We conclude by again thanking the reviewer for their praise and constructive feedback!  Please do not hesitate to reach out if you have further questions we can answer!
>
>
>
> -- Submission 14345 Authors

---

### Official Review · Reviewer_JTUb · 2025-10-31

**Soundness:** 4
**Presentation:** 3
**Contribution:** 4
**Rating:** 8
**Confidence:** 3

**Summary:**

While neural network–based adaptations of the classical Hawkes process show strong performance, they often lack interpretability.
This paper introduces an interpretable TPP model called the Hyper Hawkes Process (HHP), where interpretability is achieved through the model’s architecture design. Here, interpretability means that the model can identify the attribution or influence of each event.
The main contribution of this work is addressing the typical trade-off between interpretability and performance. The proposed method strengthens the model’s capability by increasing the latent dimension. Quantitatively, it performs competitively with baseline methods while uniquely offering interpretability.

**Strengths:**

- Adapting neural networks to classical methods improves expressiveness but can make the model more of a “black box.” The authors address this by using eigenvector decomposition to keep the model interpretable, offering a clear particle-based view and attribution.
- They also overcome the common trade-off between performance and interpretability. By increasing the latent dimension, the model becomes more expressive and reduces this trade-off.
- One interesting observation in this work is that the conventional Hawkes process ties the latent dimensions to the mark space, which opens up a promising direction for future research.

**Weaknesses:**

- Minor. The visualization of particle attribution in Figure 2 is difficult to interpret. Distinguishing sample lines using a dotted style or alternative markers could improve clarity.
- The model increases the latent dimension and adds more architectural components, which may lead to longer runtime. It is unclear whether the runtime is comparable to baseline methods.

**Questions:**

- In Figure 2, could the particle attribution be visualized more clearly, for example by using dotted lines or alternative markers to distinguish sample lines?
- Given that the model increases the latent dimension and adds more architectural components, how does its runtime compare to the baseline methods?
- Other than visualization, are there any practical applications of the particle attribution?

---

> ### Author Response · Authors · 2025-11-22
>
> Thank you to the reviewer for taking time to review our paper.  We are very pleased to see our interpretability and model architecture resonate so strongly with the reviewer!  To address the weaknesses and questions raised:
>
> - `Minor. The visualization...` & `In Figure 2...`:  Thank you for highlighting this.  We were not not overly happy with the presentation ourselves, partly due to the space constraints. We plan (/hope!) to use the extra allowed page in a camera ready to expand the visualizations for these interpretability experiments. We will certainly make the suggested changes to the line styles as part of that expansion.
>
> - `The model increases...` & `Given that the...`:  We also refer the reader to the [global response](https://openreview.net/forum?id=ci4PveaHtr&noteId=qj4xrC0DEl). The latent dimension is increased, but the computation is highly amenable to parallelization: projection operations can be efficiently parallelized across sequence length; the linear Hawkes process can be computed in parallel using a parallel scan; and the dynamics are diagonalized and hence are $O(N)$ work.  The limiting factor is therefore the actually the GRU!  As a result, we found runtimes to be comparable between the IFTPP and the HHP.  It is straightforward to substitute an attention-based architecture if runtime is paramount.  We will add discussion and comparison of this to a camera ready version.
>
> - `Other than visualization...`:  This is a _great_ question, and is something we could have made clearer.  Particle-based attribution is what facilitates both event-level attribution within a history-dependent model _and_ the leave-n-out estimators we construct.  The lack of a particle interpretation is what makes event-level interpretation in other neural MTPPs so challenging, because the nonlinearities make this attribtion impossible.  One could construct a leave-n-out estimator in neural MTPPs, but it would require re-running the model for every set of n events left out, incurring combinatorial complexity for the group-level attribution explored in "Group Influences" (near Line 393).  We will reinforce this critical link in a camera ready version.
>
>
>
> We again thank the reviewer for their positive feedback and insightful questions.  Please do not hesitate to reach out if you have further questions!
>
>
>
> -- Submission 14345 Authors

---

### Official Review · Reviewer_Lb8M · 2025-11-01

**Soundness:** 2
**Presentation:** 2
**Contribution:** 1
**Rating:** 4
**Confidence:** 4

**Summary:**

The paper models the dynamics of the Hawkes process with exponential decay kernel in a single-layer latent space and applies an affine transformation to obtain the intensity function. Further, the paper leverages a hypernetwork to infer a time-varying matrix for the exponential decay rate parameter given historical events. Model parameters are learned via maximum likelihood estimation derived from the projected intensity function. Experimental results on seven datasets for next-event prediction demonstrate log-likelihood performance competitive with baseline methods. Additionally, the paper explores the interpretability of the proposed approach by adopting event-level attribution-based techniques.

**Strengths:**

- The paper is well-written and easy to follow.
- Results on seven datasets for next-event prediction demonstrate log-likelihood performance that is competitive with baseline methods.
- The paper explores the interpretability of the proposed approach by adopting event-level attribution techniques.

**Weaknesses:**

- The paper appears to be a straightforward extension of the previously proposed DLHP approach with minimal modifications, namely, using a single-layer latent space and a hypernetwork to estimate event-specific decay rates.

- Given the limited technical contributions, the experimental results are underwhelming:

1) Although the paper claims state-of-the-art performance, the reported log-likelihood appears comparable to baseline methods.
2) Table 1: It is unclear why raw event accuracy metrics are omitted in favor of rank-based evaluations.

-  The paper asserts that the proposed approach is more interpretable than prior methods. However, by modeling dynamics in the latent space, the event-triggering kernel becomes less interpretable. It is unclear how the proposed attribution method enables recovery of the ground-truth triggering kernel. I encourage the authors to discuss this further and include qualitative results, as well as comparisons with other interpretable approaches such as [1,2]. Unfortunately, the results presented in Figure 2 are unconvincing without direct comparisons between predicted and empirical ground-truth kernels.

- The proposed approach also appears computationally expensive relative to alternatives. I encourage the authors to provide an analysis of computational complexity, including the number of parameters and training/inference time.


- Given that the ablation study shows removing the latent space and hypernetwork does not result in a significant performance drop, it is unclear why the added complexity is necessary, especially since a simpler model would be more interpretable and computationally efficient than the proposed approach.

**References**
- [1] Isik Yamac et al., "Hawkes Process with Flexible Triggering Kernels", MLHC 2023.
- [2] Pan Zhimeng et al., "Self-adaptable point processes with nonparametric time decays", NeurIPS 2021.

**Questions:**

- Table 1: Could you provide raw accuracy metrics instead of ranks?
- Tables 1–2: Could you include standard errors for the predictions?
- Could you benchmark the proposed approach against baselines in terms of computational efficiency?
- Table 1: Could you also include the ablation models?

---

> ### Author Response · Authors · 2025-11-22
>
> Firstly we thank the reviewer for taking the time to review our paper, and for praising the clarity of our submission and for highlighting the performance of our model!  We will respond to the specific weaknesses and questions raised:
>
> - `The paper...`:  We believe this is an understatement of our contribution.  While there are similarities between our architecture and DLHP, there are substantial differences in the objectives, methodology and evaluation.  We also show that a simple GRU is as performant as a deep stack of time-dependent layers, as is used in DLHP.  We also explicitly design an architecture that admits a particle-based representation, not explored or mentioned in DLHP.  We then leverage this to explore interpretability and efficient leave-n-out estimators, this is not explored or even mentioned in DLHP (in fact, DLHP explicitly highlights a lack of interpretability as a weakness of the architecture).  We also explore the impact of stateful SSMs and removal of the skip connections to aid interpretability, neither of which are not explored or even mentioned in DLHP.  Therefore, we believe the assertion that it is a “straightforward extension” is not representative of the aims nor findings of our submission.
> - `Although the...`:  We believe the reviewer has misunderstood our claim.  We argue that by matching state-of-the-art results, we are equivalent to state-of-the-art.  Note we do not claim that sole-ownership of state-of-the-art, or that we are _the_ state-of-the-art.  As evidence for this, in our contribution bullets we explicitly say: ``... HHP achieves state-of-the-art or near state-of-the-art performance on real-world MTPP benchmarks.''.  We are happy to update the wording to clarify this.
> - `Table 1:...`:  Raw numerics for log likelihoods, accuracies and calibrations **are** included in Tables 2, 5 and 6.  Table 1 is intended to be a compact summary of the results throughout the paper, which total almost three pages of tables.  We have added references to the other tables and added explanation of this in the caption for Table 1.
> - `The paper...`:  Again, we believe the reviewer has misunderstood our claims.  We explicitly **do not claim** that the HHP recovers the "ground truth triggering kernel".  In fact, we have an entire section, Section 6.3, dedicated to how the HHP _does not_ recover that kernel.  Instead, it allows us to expose and meaningfully interrogate how the model produced predictions with clear links to events.  We will also add discussion and comparison to the linked papers, as well as the ground-truth kernels to the figures, in any camera ready version.
> - `The proposed...` & `Could you...`:  Please see the [global response](https://openreview.net/forum?id=ci4PveaHtr&noteId=qj4xrC0DEl).
> - `Given that...`:  We again do not agree with the reviewers comments, and believe there is evidence to the contrary in our original submission.  Table 2 clearly shows that performance drops with each ablation made (see e.g. the drops in aggregate rank from 6.0, 4.0, 2.6 to 2.0 in Table 2 for likelihood rank).  We also state, in no uncertain terms: ``... the statefulness of the HHP does have a marked impact on performance...''.  The reviewer may be confused, as we highlight in that passage that even our ablations outperform many existing neural MTPP models (e.g. THP).  We ask the reviewer either reiterate their claims (and point to the passages of the paper that cause concern), or remove their critique.
> - `Tables 1–2:...`:  Please see the [global response](https://openreview.net/forum?id=ci4PveaHtr&noteId=qj4xrC0DEl).  We have added multiple random repeats, and observed the results were robust, with the ranks (both within metrics and in aggregate) remaining broadly unchanged.
> - `Table 1: Could...`:  Space permitting, we are happy to include the ablations in Table 1.
>
> Based on the above evidence-backed responses to all of your points, we conclude by kindly asking the reviewer to reconsider their scores for "Soundness", "Presentation", "Contribution" and "Rating":
> - Soundness:  The reviewer raised no comments pertaining to a lack of soundness of our method?
> - Presentation:  The reviewer explicitly highlighted the quality of our writing, and all extra information the reviewer asked for is either already in the paper or has been added/will be easily added.
> - Contribution:  We respectfully disagree, on the evidence above, that our contribution is "Poor":  We present a new model that is on-par with (and often better than) recent state-of-the-art models; we add interpretability and explore representations to a new family of MTPP models; we are the first paper to explore leave-one-out estimators in MTPPs; and we explore the impact of statefulness on predictions.
> - Rating:  Based on our rebuttal to all of the points the reviewer raised, we kindly ask if they would consider raising their rating.
>
> Please do not hesitate to reach out if you have further questions!
>
> -- Submission 14345 Authors

---

### Official Review · Reviewer_gtbq · 2025-11-01

**Soundness:** 3
**Presentation:** 3
**Contribution:** 3
**Rating:** 4
**Confidence:** 4

**Summary:**

The authors propose hyper Hawkes process (HHP), which uses a hypernetwork to adapt dynamics over time and to decouple its latent space from the dimensionality of the marks. Experiments demonstrate that HHP outperforms state-of-the-art models and enable insight into the model mechanics for interpretability purposes.

**Strengths:**

The proposed model is well justified and described in enough detail in the main paper to give a reader a sense of the mechanism driving the model, and unlike existing models that use a representation model for the history of events, the proposed model does it in a manner that is interpretable, i.e., the contribution of individual marks to the estimated intensities can be also estimated. Moreover, as the authors point out, the "transition" operator is both expressive and efficient due to the use of an spectral decomposition.

The leave-one-out estimator in (9), which stems from (8) constitutes a natural way to quantify the influence of an event in the history on the estimated intensity, which aids interpretability.

The experiments are, in general terms, extensive. The authors consider seven datasets, event and mark metrics spanning model likelihood, error (RMSE and accuracy) and calibration (PCE and ECE), and ablation experiments to demonstrate the contribution of each component of the proposed model. There is also an experiment illustrating the interpretation capabilities of the model.

**Weaknesses:**

The main weakness of the proposed model lies on the experimental evaluation of the proposed model. Specifically, the proposed model has the overall best rank (Table 1), but is only better than the competing methods in half of the metrics. However, the bigger issue is that the metrics do not account for variation, thus it is very difficult to assess the significance of the results, so in that sense it may be possible that the difference between HHP and DHLP is not at all significant. Moreover, the poor calibration performance of the model is concerning in the sense that it greatly impacts interpretability, which is a core value of the proposed model.

Although presented as an advantage, the results indicate that the expressivity of a time-dependent transition operator may not be as useful at least in terms of performance.

The interpretability experiment is a welcome addition, however, it will be better if the authors also presented interpretability experiments with real data. On the same line, although the interpretability of the model is a desirable property, one wonders if interpretation at the particle level is realistic in practical scenarios.

**Questions:**

- Figure 1 needs either a better caption or needs to be described in more detail.
- Have the authors consider an ablation where beta_t = D_i? which seems related to the ablation study where V_i = V.
- How to reconcile the poor calibration with the better likelihood metrics?

---

> ### Author Response · Authors · 2025-11-22
>
> Thank you to the reviewer for taking time to review our paper.  Thank you for your comments on the clarity of our exposition, and for describing our experiments as ``extensive''. We refer you to the [global response](https://openreview.net/forum?id=ci4PveaHtr&noteId=qj4xrC0DEl) for responses that affect multiple reviewers.  We respond to your more fine-grained response here:
>
> - `The main...`:  We have added multiple repeats for all models and metrics.  The final rankings and findings are broadly unchanged.  Please see the updated paper for these updated results and global response around the result and state-of-the-art claims.
>
> - `Moreover, the...` & `How to...`:  Please see the [global response](https://openreview.net/forum?id=ci4PveaHtr&noteId=qj4xrC0DEl).  On reconciling, crucially, calibration measures something very different from the other prediction metrics, and is a very nuanced metric (e.g. the marginal distribution is perfectly calibrated but is a poor predictor), and so calibration should always be considered alongside predictive performance, and better predictive models may be slightly worse calibrated than worse predictive models, but still generate better predictions overall. We have added discussion of this.
>
> - `Although presented...`:  We may have misunderstood the reviewers query and would appreciate clarification:  Table 2 clearly shows that ablating any part of the HHP is detrimental to performance (see e.g. the drops in aggregate rank from 6.0 ($\lnot$Latent) to 4.0 ($\lnot$Hyper), 2.6 ($\lnot$Stateful) to 2.0 (full model) in Table 2).  Crucially, the $\lnot$Stateful ablation does still has time-varying _diagonal_ dynamics, it just has a fixed eigenbasis in which those dynamics evolve ($V_i$ is fixed, but $D_i$ is time-varying).  The ablations $\lnot$Latent and $\lnot$Hyper then have fixed temporal dynamics, and both suffer a large performance degradation.  In our understanding, this clearly indicates that time-dependence is a huge performance booster.  We kindly ask the reviewer to restate their concern, and where in the paper the cause for concern is.
>
> - `The interpretability...`:  Please see the [global response](https://openreview.net/forum?id=ci4PveaHtr&noteId=qj4xrC0DEl).
>
> - `On the...`:  This is a great question.  We contend that the particle representation is _more_ efficient and practical:  The number of particles required is **equal** to the number of events, and subsequent investigations and manipulations (e.g. the leave-one-out estimators) are computationally efficient (i.e. the linear Hawkes can be computed in parallel, the recurrence parameters do not need to be re-computed, the particles are additive etc.).  This is in stark contrast to other black-box neural MTPP methods, where the entire model must be re-run for every combination of events being studied, resulting in combinatorial complexity for the ``Group Influence'' experiments (see Section 6.1, Figure 9).  We therefore argue that actually the particle-based is **more** practical in real-world scenarios!  That you for raising this excellent point, we have added discussion of this to the revised paper.
>
> - `Figure 1...`:  We have updated the figure & caption, thank you for catching this.
>
> - `Have the...`:  Ahh, yes, this is a very subtle point.  The case where $\beta_t := D_i$ is actually _equivilant_ to when $V_i = V$, which can be seen by $V_i = \mathbb{I}$ (the identity matrix), which is _constant_, analogously to $V_i = V$.  Because $B$ and $C$ are unconstrained, any $V$ can be recovered through an appropriate fixed (i.e. not stateful) rotation.  This case is therefore denoted as HHP\textsubscript{$\lnot$Stateful} in Table 2!  We will add highlighting of this equivilance to the text, thank you for highlighting it!
>
> We conclude by again thanking the reviewer for catching some oversights on our part, and for highlighting some places where we can improve clarity!  We ask that, given the reviewers numeric Soundess, Presentation and Contribution scores, and our response to their initial comments, if the reviewer would consider upgrading their overall rating?  We welcome any further questions you may have!
>
> -- Submission 14345 Authors

---

### Author Response · Authors · 2025-11-22
**Global Response**

Firstly, we thank all four reviewers for taking the time to read and review our submission; we were thrilled that the reviewers highlighted so many positive aspects of our work and had some constructive feedback.  We presented the hyper Hawkes process (HHP), a novel marked temporal point process architecture that uses a hyper network to predict the time-varying and data-dependent parameters of a generalized linear Hawkes process. We show that this architecture achieves both state-of-the-art performance (or at the very least in-line with state-of-the-art), while also exposing internal computational mechanisms, allowing us to define novel interpretability mechanisms by making a connection with leave-one-out estimators.

There were a couple of piece of feedback that were highlighted by multiple reviewers, we will address them here, before responding individually:

- Reviewers gtbq and Lb8M note that we did not include multiple random seeds.  We have included multiple seeds for all models and methods in the updated paper (Retweet and LastFM are still running).  **The rankings of the models are broadly unchanged**, indicating that the performance of the HHP is robust to initialization.

- Reviewers also raised questions about the computational complexity (Lb8M and JTUb) and parameter complexity (Lb8M). Notably, HHP achieves this level of performance while using, on average, 54% fewer parameters than S2P2 across datasets. The HHP matches IFTPPs and NHPs $O(L)$ work and computational depth, noting that HHP and IFTPP are limited by the highly optimized GRU, and NHP is limited by the slower T-LSTM.  This is compared to the $O(\log(L))$ depth of attention-based models but their $O(L^2)$ work.  This theoretical difference is borne out in practice, with HHP being: about half the throughput of IFTPP; an order of magnitude higher throughput than NHP; and slower than attention-based models, but is able to scale to much longer sequences.  We will add a full table of parameter counts for each of the models and variants, and a table of complexity and runtime comparisons to any camera ready version.

Reviewer gtbq also commented on the calibration. We first note that calibration is a highly nuanced metric that requires careful handling (see Bosser & Taieb). For instance the marginal distribution is perfectly calibrated, but is obviously a poor predictive model. Therefore, other metrics should always be taken into account when considering calibration, and being well-calibrated does not mean you are necessarily a better predictive model. Finally, looking at the average calibration in Appendix Table 6, we see that all models are comparably calibrated, with no model being a stand-out winner or loser.

With this in mind, some reviewers also referred to whether the results qualified as “state of the art”.  We highlight that: (a) Our paper is not chasing absolute expressivity or predictive performance, but instead a model that is, at the _very least_, comparable with state of the art while adding interpretable aspects. We believe on the evidence, we have achieved this. (b) Secondly, of the four predictive metrics measured (time and mark log likelihood, and time and mark accuracy), HHP is ranked first for three of them, and second for the other.  Unlike predictive metrics, an average calibration _can_ be computed across different datasets so we do not need to resort to ranking. Inspecting the average calibration scores in Appendix Table 6, we see that all models are similarly calibrated on average, with S2P2 edging out baselines in PCE, and being mid-table in ECE.  We therefore believe, on this evidence and the strong and balanced aggregate performance, that the HHP is at least _comparable_ with state-of-the-art.  We will add clarification of this reasoning to any camera ready.

We will now respond in more detail to individual reviewer comments.  Thank you again for reviewing our paper and providing both positive and constructive feedback!  Please do not hesitate to reach out if you have further questions!

Thank you very much!

-- Submission 14345 Authors

---

### Author Response · Authors · 2025-12-03
**Finalized Results**

We have uploaded a revised manuscript with multiple seeds for all main models and baselines.  We see that, averaged across datasets and metrics, HHP matches the aggregate performance of the SotA S2P2 architecture, highlighting HHPs excellent and robust predictive performance, all while using 54% fewer parameters (on average) and providing interpretable insights.

We again thank all four reviewers and both ACs for their time and attention reviewing our paper!

-- Submission 14345 Authors

---

### Meta-Review · Area_Chair_rrWG · 2026-01-05

**Summary:**

The reviewers consistently question the limited novelty, noting that the method is a modest extension of existing DLHP-style models with unclear benefits from the added latent space and hypernetwork. They also highlight insufficient experimental rigor, including missing variance estimates, reliance on rank-based metrics, lack of efficiency analysis, and weak ablation support. Finally, the interpretability claims are not well substantiated, as evaluations are limited to synthetic data and lack convincing comparisons with prior interpretable Hawkes models. Therefore, I recommend rejection.

**Reviewer Concerns:**

I think the concerns raised by Reviewer gtbq, Reviewer Lb8M, and Reviewer gUR2 have not been fully addressed.

**Reviewer Scores:**

I think Reviewer gtbq, Reviewer Lb8M, and Reviewer gUR2 might collectively increase their scores by 0–2 points if they were able to fully participate in the discussion.

---

### Decision · Program_Chairs · 2026-01-26

Reject